# Regional assessment of extreme sea levels and associated coastal flooding along the German Baltic Sea coast

Joshua Kiesel[1], Marvin Lorenz[2], Marcel König[3], Ulf Gräwe[2], and Athanasios T. Vafeidis[1]

[1]Department of Geography, Christian-Albrechts-Universität zu Kiel, 24118, Germany
[2]Leibniz Institute for Baltic Sea Research Warnemünde, Rostock, 18119, Germany
[3]Private Consultant, now at Center for Global Discovery and Conservation Science, Arizona State University, Tempe, AZ 85281, USA

**Correspondence:** Joshua Kiesel (kiesel@geographie.uni-kiel.de)

**Abstract.**

Among the Baltic Sea littoral states, Germany is anticipated to endure considerable damage as a result of increased coastal flooding due to sea-level rise (SLR). Consequently, there is a growing demand for flood risk assessments, particularly at regional scales, which will improve the understanding of the impacts of SLR and assist adaptation planning.

Existing studies on coastal flooding along the German Baltic Sea coast have either used state-of-the-art hydrodynamic models, but cover only a small fraction of the study region, or assess potential flood extents for the entire region, but rely on global topographic data sources and apply the simplified bathtub approach. In addition, the validation of produced flood extents is often not provided.

Here we apply a fully validated hydrodynamic modeling framework covering the German Baltic Sea coast that includes the height of natural and anthropogenic coastal protection structures in the study region. Using this modeling framework, we extrapolate spatially explicit 200-year return water levels, which align with the design standard of state embankments in the region, and simulate associated coastal flooding. In specific, we explore; 1) how flood extents may change until 2100 if dike heights are not upgraded, by applying two high-end SLR scenarios (1 m and 1.5 m); 2) hotspots of coastal flooding, and; 3) evaluate the use of SAR-imagery for validating the simulated flood extents.

Our results confirm that the German Baltic coast is exposed to coastal flooding, with flood extent varying between 217 km$^2$ and 1016 km$^2$ for the 200-year event and a 200-year event and 1.5 m SLR, respectively. Most of the flooding occurs in the federal state of Mecklenburg-Western Pomerania, while extreme water levels are generally higher in Schleswig-Holstein. Our results emphasise the importance of current plans to update coastal protection schemes along the German Baltic Sea coast over the 21$^{st}$ century in order to prevent large-scale damage in the future. Our results confirm that the German Baltic coast is exposed to coastal flooding, with flood extent varying between 217 km$^2$ and 1016 km$^2$ for the 200-year event and a 200-year event and 1.5 m SLR, respectively. Most of the flooding occurs in the federal state of Mecklenburg-Western Pomerania, while extreme water levels are generally higher in Schleswig-Holstein. Our results emphasise the importance of current plans to update coastal protection schemes along the German Baltic Sea coast over the 21$^{st}$ century in order to prevent large-scale damage in the future.

## 1 Introduction

Sea-level rise (SLR) will increase the frequency and intensity of extreme water levels thus increasing the risk of coastal flooding with potentially far-reaching socio-economic impacts (Kirezci et al., 2020; Hinkel et al., 2014). In the European Union, 100,000 people are already exposed to floods every year, and the average annual flood damages are estimated to 1.4 billion €. These
numbers are likely to increase by the end of the century (Vousdoukas et al., 2020). Across Europe, the Baltic Sea and the North Sea are projected to experience the highest increase in extreme sea levels (ESL) towards the end of the century (Vousdoukas et al., 2017). In the Baltic Earth Assessment Reports (BEAR), Rutgersson et al. (2022) suggest that Germany, among other Baltic Sea littoral states, is likely to suffer severe damage from increased coastal flooding due to climate change.

In the Baltic Sea, ESLs occur on various spatio-temporal scales and all phenomena contributing to ESL are mainly generated
by meteorological and, to a limited extent, astronomical factors (Weisse and Hünicke, 2019). The most important contributions to ESLs come from storm surges, wind waves, and preconditioning, the latter of which leads to increased water volumes in the Baltic Sea before the onset of a storm (Weisse et al., 2021; Suursaar et al., 2006; Madsen et al., 2015). Due to the microtidal regime, storm surges can persist for relatively long periods, ranging from hours to several days (Wolski and Wiśniewski, 2020; MacPherson et al., 2019).

Facing a growing demand for flood risk assessments at regional scales, a better understanding of the consequences of natural hazards such as storm surges and associated extreme sea levels (ESL) is required. Such assessments can enable society and decision makers to selectively invest in adaptation options and support policy-making (Vousdoukas et al., 2018; Rutgersson et al., 2022). The key component in improving our understanding of the impacts of SLR and ESL constitutes the detailed
mapping of coastal floodplains. High-quality inundation maps, including information on flood depth and extent, are critical for coastal impact assessments, since the density of valuable assets often tends to increase towards the coast (Vousdoukas et al., 2016). This need has also been identified by European legislation: since 2007, the EU Flood Directive (Directive 2007/60/EC) requires member states to identify and map areas, assets and populations at risk from flooding along coastlines and water courses, and to take measures to reduce this risk. For this purpose, the Directive obliges member states to produce high quality
inundation maps that account for ESL, SLR and existing coastal protection infrastructure.

State-of-the-art coastal flood maps should consider oceanographic forcings, projected SLR, detailed topographic data including anthropogenic coastal protection measures such as dikes, and the effects of land cover on flood propagation. Oceanographic forcings determining flood characteristics are the magnitude of the surge (i.e. the peak water level), the duration of the surge (particularly in the Baltic Sea), and wave setup (Höffken et al., 2020; Hendry et al., 2019). Topographic data sets constitute a
major source of uncertainty in large-scale coastal flood modeling, particularly because natural and anthropogenic flood barriers such as dikes are often not sufficiently well resolved (Leszczyńska et al., 2022; Vousdoukas et al., 2018). Problematically, disregarding human adaptation constitutes the largest bias in regional to global scale flood risk assessments and data availability

is often a major concern (Hinkel et al., 2021). Even if data of good quality exists for a specific region, the simulation of coastal flood extents at a regional scale is computationally expensive, often leading to the application of simplified, static bathtub models in coastal flood impact assessments. The bathtub model maps all coastal areas hydrologically connected to the sea below a certain (extreme) water level as inundated, without accounting for the alteration of flow as a consequence of varying surface roughness or the temporal evolution of the surge (e.g. duration). Even though these models may perform well depending on the geomorphic setting (Kumbier et al., 2017), several recent studies have pointed out that the bathtub model can overestimate coastal flood extents (Vousdoukas et al., 2016; Lopes et al., 2022; Didier et al., 2019).

Due to the drawbacks of static inundation models, more complex, hydrodynamic models are increasingly used to map coastal floodplains from local to continental scales (Vousdoukas et al., 2016; Lopes et al., 2022; Didier et al., 2019; Bates et al., 2021; Leijnse et al., 2021). However, the application of such models is complex, sensitive to the model setup (surface roughness parameterisation, solvers etc.) and computationally demanding, particularly when modelling at higher resolutions. This constitutes a dilemma, as the quality of regional scale assessments can be affected by coarse resolutions, which result in inaccuracies associated with the representation of natural and anthropogenic flood barriers such as beach ridges, dunes, seawalls or dikes. In addition, comprehensive datasets on the location and characteristics of coastal protection infrastructure are missing for many parts of the world. In this context, Vousdoukas et al. (2018) highlight the urgent need for broad-scale but highly detailed datasets of coastal protection standards. Additional uncertainties arise from the lack of data to validate modelled flood extents, the extrapolation of return water levels beyond the length of tide gauge records and the projections of future sea levels (Vousdoukas et al., 2016, 2018).

Model validation is one of the major challenges in coastal flood risk analysis as it provides an awareness of model limitations. Recent advances in the application of hydrodynamic models for flood impact assessments have not been matched by advances in model validation (Rollason et al., 2018); in particular, the lack of validation data of appropriate spatial and temporal coverage constitutes a major problem (Molinari et al., 2019). For instance, studies that have modelled on global to continental scales have validated simulated flood extents with data covering only a very small part of the entire study region (Vousdoukas et al., 2016; Sampson et al., 2015).

Along the German Baltic Sea coast, existing studies on coastal flooding have either used state-of-the-art hydrodynamic models, but cover only a small fraction of the study region (Höffken et al., 2020; Vollstedt et al., 2021), or assess potential flood extents for the entire region, but rely on global topographic data sources and apply the bathtub approach (Schuldt et al., 2020). In addition, the validation of produced flood extents is not provided. There is a need to simulate coastal flooding on a regional scale, considering the limitations of large-scale coastal flood mapping mentioned before. This is particularly true for limitations associated with topographic data sources and the incorporation of coastal protection infrastructure, which constitute the main bottlenecks for the quality of coastal flood risk assessments (Vousdoukas et al., 2018; Hinkel et al., 2021).

The German Baltic Sea coast comprises a total length of 2538 km and 27 % of this coast is protected by dikes (Sterr, 2008; van der Pol et al., 2021). Today, state embankments in both federal states, Schleswig-Holstein (SH) and Mecklenburg-Western Pomerania (MP) are designed to be high enough to prevent flooding during a storm surge with a return period of 200 years

plus a buffer of 0.5 m to account for SLR (Melund, 2022; StALU, 2012). On the other hand, regional dikes have a variable but generally lower standard of protection (Melund, 2022). During the last two decades, the concept of so-called *climate dikes*

has been introduced as a paradigm in embankment construction. The *climate dike* accounts for uncertainties related to SLR projections by having wider dike crests, which allows for a comparatively easy and rapid increase in dike heights without reconstructing the dike base. The *climate dike* can easily be increased in height by up to 1.5 m, and suggested extension options include 0.5 m and 1.0 m (Melund, 2022).

Here, we simulate coastal flooding along the German Baltic Sea coast for a storm surge that aligns with the design standard of state embankments in the region, i.e. the 200-year return water level. This study aims at; 1) exploring how the flood extent may change until the end of the century, if existing dikes are not upgraded, by applying two high-end SLR scenarios (1 m and 1.5 m); 2) identifying hotspots of coastal flooding in the study region, and; 3) evaluating the use of SAR-imagery for validating the simulated flood extents. To the knowledge of the authors, this study constitutes the first regional-scale assess-

ment using a high resolution, fully validated, and offline-coupled hydrodynamic modelling framework that incorporates natural and anthropogenic flood barriers to assess extreme sea levels and associated coastal flooding along the German Baltic Sea coast.

We apply a newly developed hydrodynamic modelling framework for the entire German Baltic Sea coast. Specifically, we use high-resolution topographic data sources extracted from a 1 m x 1 m Light Detection and Ranging (LiDAR) derived

Digital Elevation Model (DEM) to identify the height of natural and anthropogenic coastal protection structures such as dunes and dikes. Further, we have access to detailed data on the location of all dikes in the study region. We consider the temporal evolution of storm surges, the effects of spatially varying surface roughness on flood propagation, and explore the suitability of Sentinel-1 Synthetic Aperture Radar (SAR) data to validate a large fraction of our study region. The applied modeling framework offline-couples a hydrodynamic coastal inundation model (50 m resolution) of the German Baltic Sea coast with

a hydrodynamic coastal ocean model (200 m resolution) covering the entire western Baltic Sea. The coastal ocean model produces spatially varying boundary conditions for the coastal inundation model. Using this setup, we perform four storm surge simulations representing 1) a 200-year event, 2) a 200-year event and 1 m SLR, 3) a 200-year event and 1.5 m SLR, and 4) a simulation of the storm surge that occurred in the study region on 2[nd] January 2019. We use the 2019 storm surge to validate the water levels generated by the coastal ocean model and the flood extent generated by the coastal inundation model.

We compare flood maps derived from our coastal inundation model with flood extents generated from SAR satellite imagery. These images were acquired on the same day of the surge, only a few hours after it peaked. More importantly, they cover a significant part of the study area. Since local tide gauges have not yet recorded a storm surge with a 200-year return period, we constructed these events by using extreme value statistics and extracting mean surge shapes from a hindcast simulation of the coastal ocean model.

## 2 Study area

The Baltic Sea constitutes a semi-enclosed, brackish water basin that is comparatively shallow (between 53 m and 55 m on average depending on dataset (Jakobsson et al., 2019)) and has its only connection to the North Sea through the Kattegat and Skagerrak (Fig. 1). The Baltic Sea is characterised by a microtidal regime (tidal range varying between 0.1 m and 0.2 m (Sterr, 2008)), low salinity, strong stratification, and anoxic conditions in many areas (Meier et al., 2022). The German part of the Baltic Sea is located in the south-west of the sea´s catchment and comprises the federal states of Schleswig-Holstein (SH) and Mecklenburg-Western Pomerania (MP) (Fig. 1). The coastal length of the German Baltic Sea is approximately 2538 km, of which 649 km are located in SH and 1889 km in MP (van der Pol et al., 2021). The German Baltic Sea coast is characterised by fjords, lagoons, islands, beaches, and soft cliffs. In contrast to the emerging northern Baltic Sea coast, parts of the southern Baltic Sea coast are subsiding as a consequence of glacial isostatic adjustment. While subsidence is generally variable and mostly well below 1 mm yr$^{-1}$, it can locally reach up to 2 mm yr$^{-1}$ (Richter et al., 2012; Weisse et al., 2021; Dangendorf et al., 2022). Due to the spatial variability, limited consistent information and rates mostly well below SLR, we have excluded subsidence from the present analysis.

The German Baltic Sea experiences storm surges mostly during strong easterly winds. The highest surge to date occurred in 1872, reaching peak water levels between 2.4 and 3.4 m above the German Ordnance Datum (NHN) in the federal state of SH. In this event, 31 people died and 15,000 lost their homes. The 1872 surge was a turning point for adaptation planning along the German Baltic Sea coast, resulting in new design standards for embankment constructions in the aftermath of the storm (Hofstede and Hamann, 2022).

## 3 Methods and data

### 3.1 Overview of modelling framework and simulated scenarios

In order to simulate flood characteristics along the German Baltic Sea coast, we employ a new modelling framework where we offline-couple two hydrodynamic models, a coastal ocean model (GETM) with a coastal inundation model (LISFLOOD-FP). The coastal ocean model provides boundary conditions, which we use to simulate coastal flood characteristics in the inundation model. We apply this model setup to simulate four events and scenarios: 1) The storm surge that occurred on 2$^{nd}$ January 2019, which we use to validate the coastal ocean and the inundation model. 2) A 200-year event that is used to determine the design heights of dikes along the German Baltic Sea coast. 3) The 200-year event and 1 m of SLR and 4) a high-end scenario of the 200-year event including 1.5 m of SLR (SLR until the year 2100). We add SLR linearly to the 200-year event (Hieronymus et al., 2018). The SLR scenarios correspond to the regional scale medium confidence projections of SSP5-8.5 (ranging between the 50$^{th}$ and 83$^{rd}$ percentiles) of the sixth assessment report of the Intergovernmental Panel on Climate Change (IPCC) for the tide gauges in Lübeck-Travemünde and Wismar (Fox-Kemper et al., 2021). Both sea-level rise scenarios are used in current coastal protection planning within the scope of the climate dike concept as so-called "building reserve". The building reserve allows the dike heights to be increased by up to 1.5 m at the end of this century with comparatively little effort (Melund, 2022).

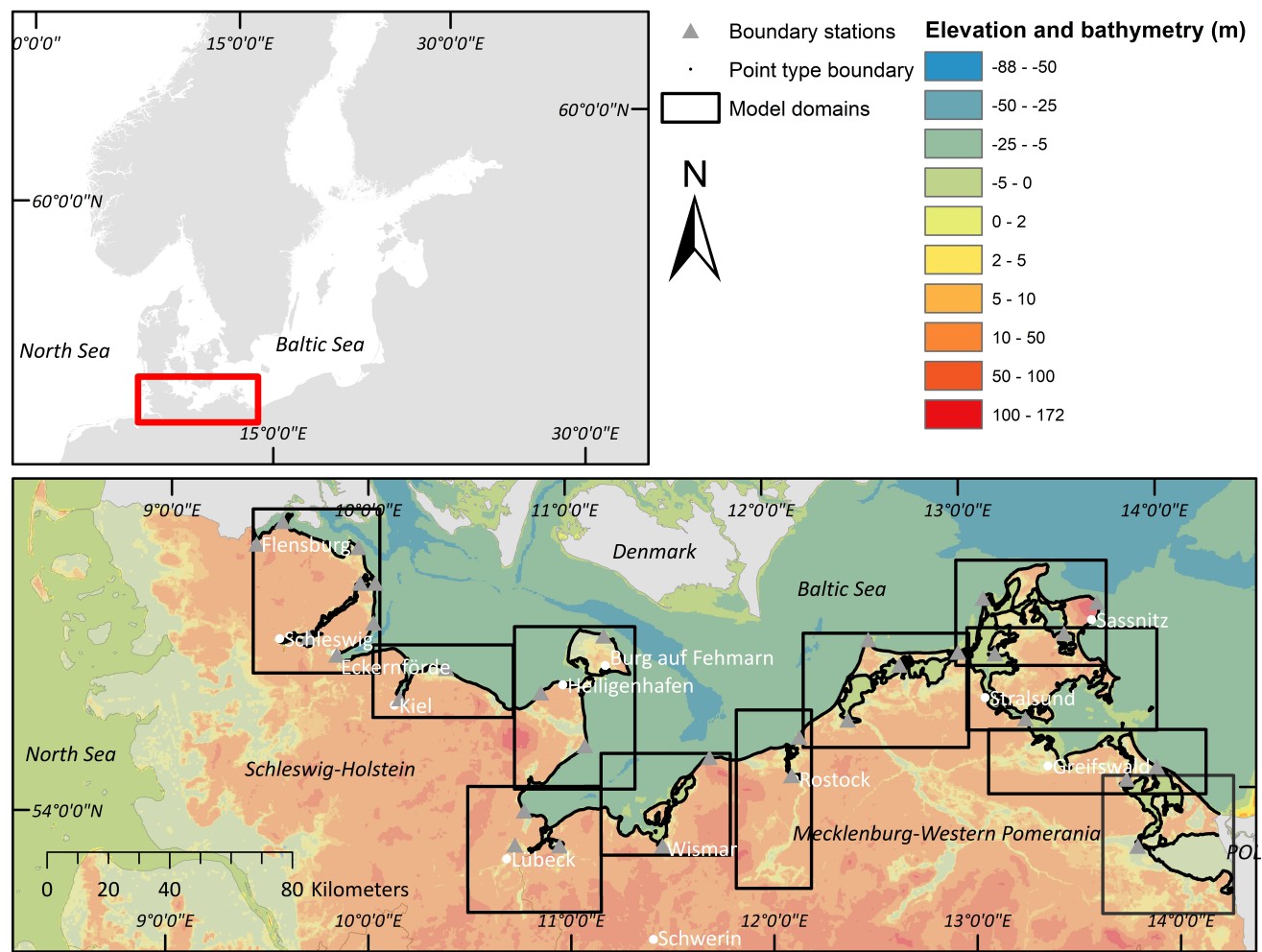

**Figure 1.** Top left: Overview map of the study region. Bottom: The German Baltic Sea and depiction of coastal inundation model setup. Bathymetry data shown in this figure was taken from the Baltic Sea Bathymetry Database (Helcom Secretariat) and the elevation data from Global multi-resolution terrain elevation data 2010: U.S. Geological Survey Open-File Report 2011-1073, 26 p. http://pubs.usgs.gov/of/2011/1073/, available via Helcom.

## 3.2 Numerical setups

### 3.2.1 Coastal Ocean Model: GETM

The General Estuarine Transport Model (GETM, Burchard and Bolding, 2002) constitutes a structured coastal ocean model
(Klingbeil et al., 2018) that solves the Reynolds averaged Navier-Stokes equations (RANS) in Boussinesq approximation. In

this study, we employ it to perform a hindcast simulation of the sea level evolution from 1961 to 2018 using its vertically integrated mode.

We employ a setup of the western Baltic Sea in 200 m resolution based on the bathymetry obtained from the European Marine Observation and Data Network (EMODnet, https://emodnet.ec.europa.eu). As atmospheric forcing we use the HARMONIE V1 dataset (https://apps.ecmwf.int/datasets/data/uerra/) of the 'Uncertainties in Ensembles of Regional ReAnalyses' (UERRA, https://www.uerra.eu) project, i.e. hourly wind (in 10 m height) and sea level pressure fields in 11 km resolution. The wind speeds of the forcing have been increased by 7% to improve the model's ability to match the observed peak water levels during the storm surges (Lorenz and Gräwe, 2023). The friction in the model is depth-dependent using the *law of the wall* with a roughness length of $z_0 = 5e - 05$ m. This value is smaller than usual but the model still tends to underestimate ESLs (see Section 4.1). The Total Variation Diminishing scheme 'Superbee' is used as the advection scheme (Pietrzak, 1998). The 200 m setup uses boundary conditions of a nesting hierarchy starting with a coarse setup of the Northwest Atlantic Ocean (see Gräwe et al. (2015) for a detailed description of the nesting). The boundary conditions are extracted from a 1 NM setup of the North and Baltic Sea, which uses ERA5 (Hersbach et al., 2020) as atmospheric forcing (winds and pressure, hourly, 32 km resolution), but neglecting tides for this study as a consequence of the microtidal regime of the study region (Gräwe and Burchard, 2012; Arns et al., 2020). Since the UERRA data ended in 2018, the 2019 surge used here to validate the simulated flood extent is computed using the German Weather Service (DWD) forecast (3-hourly, 7 km; Zängl et al., 2015). In order to keep the computation of the modelled return levels consistent to one dataset, the 2019 surge is not included in the return level computation.

### 3.2.2 Coastal inundation model: LISFLOOD-FP

LISFLOOD-FP (hereafter referred to as Lisflood) is a raster flood inundation model that is used to simulate fluvial or coastal flood propagation (Bates et al., 2013). Lisflood includes several solvers that simulate the propagation of the flood wave along channels and across floodplains using simplifications of the shallow water equations. This makes it a reduced complexity model that is a faster alternative to full shallow-water models but providing results of similar accuracy (Neal et al., 2011; Bates et al., 2013). Therefore, Lisflood is particularly useful for large-scale inundation modelling that would otherwise be too computationally expensive (Vousdoukas et al., 2016; Bates et al., 2021).

### 3.2.3 Model setup in Lisflood

From the available solvers in Lisflood, we applied the floodplain solver 'Acceleration', which is a simplified form of the shallow water equations, excluding only the convective acceleration term. The flow is calculated using 'Acceleration' as a function of friction, water slopes, and local water acceleration (see Bates et al. (2013) for the respective equations). The timestep varies throughout the simulation according to the Courant-Friedrichs-Lewy condition and is related to the cell size and water depth (Bates et al., 2013).

In order to model the entire German Baltic Sea coast in 50 m resolution, we divided the study region into eleven model domains, each covering an area of on average 1739 km$^2$ (Fig. 1). Model domains were defined considering water level vari-

ability across the study region, so that each model domain is characterised by comparatively homogeneous water levels. We used a point-type boundary to force the model, with a boundary point placed every 50 m along the model coastline (55,169 points in total) (Table 1). In order to account for spatial variations in water levels within each model domain, we defined a total of 32 'boundary stations' (Fig. 1), for which hydrographs (water level time series) were modelled in the coastal ocean model (GETM) for the four storm surge scenarios (2019 surge, 200-year event, 200-year event and 1 m and 1.5 m of SLR). The 32 boundary stations used in this study are located at different locations as compared to existing tide gauges. Each boundary point received the boundary conditions (the hydrograph used to force the coastal inundation model in Lisflood) from the nearest boundary station (Fig. 1). In cases where the nearest flood boundary station to a boundary point located on the open coast was situated inside protected fjords or lagoons, or vice versa, we manually corrected that point to ensure that open coast boundary points are not forced with hydrographs of sheltered locations.

The model elevation and bathymetry were compiled as follows: we first aggregated the 10 m LiDAR-derived elevation data to 50 m in order to make it consistent with the bathymetry data (Table 1). Then, both datasets were merged by differentiating between sea (bathymetry) and land (elevation) areas. A resolution of 50 m is insufficient to reliably resolve natural and anthropogenic coastal protection barriers, as dike crests are typically narrower than 50 m, so grid sizes smaller than 10 m are recommended (Vousdoukas et al., 2012b, a).

We incorporated detailed information on dikes (location and height) into the modeling framework by using a high-resolution LiDAR-derived (1 x 1 m) DEM and comprehensive datasets on the location of both state and regional dikes from local state authorities (Table 1). The incorporation of dikes constitutes one of the major improvements of the applied modeling framework as compared to previous regional or continental scale assessments (Vousdoukas et al., 2016). Elevation data and coastal protection levels are considered as the main bottlenecks for the quality of coastal flood risk assessments (Vousdoukas et al., 2018; Hinkel et al., 2021). Without correcting for dike heights in a 50 m DEM, the simulated flood extent will be overestimated, as the elevation of the dike heights are averaged out due to the resolution (Vousdoukas et al., 2012b, a). The difference of a DEM with and without dike height correction is shown in Appendix Fig. A1. The integration of dikes into the 50 DEM was done as follows: First, we extracted elevations from the 1 m x 1 m DEM within a 100 m buffer around the coastline and the dike shapes of SH and MP. We extracted 1 m x 1 m elevations around the coastline in order to accurately resolve natural flood barriers such as dunes and beach ridges, but to also account for a variety of hard coastal protection measures, such as revetments and seawalls. The 100 m buffer was used to account for inaccuracies related to the data on dike positions and the DEM. In the next step, we aggregated these datasets to 50 m by using maximum elevations and merged them with the elevation-bathymetry data.

**Table 1.** Datasets used to set up the coastal inundation model in Lisflood.

| Dataset | Resolution | Source | Accuracy |
|---|---|---|---|
| Bathymetry | 50 m | Federal Maritime and Hydrographic Agency (BSH) | *NA* |
| Elevation | 10 m | ATKIS® DEM 10 (LiDAR); State Office of Geoinformation, Surveying and Cadestre MP and State Office for Suveying and Geoinformation SH | 0.5 - 2.0 m |
| Elevation | 1 m | ATKIS® DEM 1 (LiDAR); State Office of Geoinformation, Surveying and Cadestre MP and State Office for Suveying and Geoinformation SH | < 30 cm horizontally and 15 - 20 cm vertically in flat terrain |
| Land cover | 100 m | Corine (© European Union, Copernicus Land Monitoring Service 2018, European Environment Agency (EEA)) | Geometric accuracy < 100 m; Thematic accuracy > 85 % |
| Coastline | shapefile | (van der Pol et al., 2021) | *NA* |
| Dikes SH | shapefile | ATKIS®; State Office for Suveying and Geoinformation SH | The dataset contains full coverage of flood protection dikes in SH. We used the layers "rel01" and "geb03". Selected shapes are "Hochwasserdeich", "Hauptdeich", "Landesschutzdeich", "Überlaufdeich", "Leitdeich", "Schlafdeich", "Mitteldeich", "Binnendeich", "Hauptdeich 1. Deichlinie" and "2. Deichlinie". |
| Dikes MP | shapefile | State Office for Agriculture and the Environment Mittleres Mecklenburg, Coastal Division (internal data) | State dikes and regional dikes of water and soil associations |

We derived the surface roughness of the study region by using land cover data from Corine (© European Union, Copernicus Land Monitoring Service 2018, European Environment Agency (EEA)) (Table 1) to assign Manning's $n$ coefficients. These coefficients are commonly employed to parameterise bottom friction of various land cover types in hydrodynamic simulations
(Garzon and Ferreira, 2016). Since Manning's $n$ coefficients are not available for all Corine classes present in the study region, we reduced the number of classes by reclassification. The reclassification scheme is provided in Table A1 in the Appendix. We then assigned five configurations of Manning's $n$ coefficients to the remaining ten categories. We followed the approach described in Höffken et al. (2020). First, we searched the literature for a variety of Manning's $n$ surface roughness coefficients for each of the ten land cover classes. We then categorised them into high, low and moderate values (Table 2). Some studies
have used uniform Manning's $n$ coefficients or setups where the only separation in surface roughness was made between land

and water areas (Table 2). We have added these two setups to the configurations of Manning´s $n$ coefficients that we used for model calibration (Table 2).

In this study, model outputs (including the validation run representing the 2019 storm surge) refer to the maximum water depth (and thus extent) predicted by the model for each pixel over the course of the simulation period.

**Table 2.** Configuration of Manning's $n$ coefficients for selected land cover classes.

| Land use class | High | Low | Moderate | Uniform | Land/water |
|---|---|---|---|---|---|
| Agriculture | 0.06 [1] | 0.03 [3,5] | 0.04 [1] | 0.035 [8] | 0.03 [2] |
| Forest | 0.2 [2] | 0.1 [4,5] | 0.15 [3, 6, 7] | 0.035 [8] | 0.03 [2] |
| Urban | 0.15 [1] | 0.015 [6] | 0.07 [3] | 0.035 [8] | 0.03 [2] |
| Wetland | 0.08 [1] | 0.035 [1] | 0.06 [1] | 0.035 [8] | 0.03 [2] |
| Sea and ocean | 0.03 [2] | 0.012 [2] | 0.02 [2] | 0.035 [8] | 0.02 [2] |
| Inland waterbodies/-courses | 0.06 [2] | 0.02 [2] | 0.035 [2] | 0.035 [8] | 0.035 [2] |
| Green urban areas | 0.12 [1] | 0.035 [1,7] | 0.07 [1] | 0.035 [8] | 0.03 [2] |
| Natural grasslands | 0.042 [1] | 0.034 [4] | 0.035 [7] | 0.035 [8] | 0.03 [2] |
| Traffic | 0.032 [1] | 0.013 [5] | 0.016 [6, 7] | 0.035 [8] | 0.03 [2] |
| Unvegetated coastal sediment | 0.09 [4] | 0.025 [6] | 0.04 [1, 3] | 0.035 [8] | 0.03 [2] |

[1](Bunya et al., 2010), [2](Garzon and Ferreira, 2016), [3](Wamsley et al., 2009), [4](Liu et al., 2013),
[5](Papaioannou et al., 2018), [6](Hossain et al., 2009), [7](Dorn et al., 2014), [8](Liang and Smith, 2015)

### 3.3 Sensitivity analysis

We tested the sensitivity of our model to variations in surface roughness coefficients using the different values of Manning's $n$ coefficients shown in Table 2. For this purpose, we modelled the storm surge from January 2019 and selected the three model domains that contained the largest flood plains (1, 7 and 8 from left to right in Fig. 1). The analysis showed that our model results are robust against variations in surface roughness coefficients (Table A2). In specific, variations in flood extent between highest and lowest Manning's $n$ coefficients vary from 0.09 km$^2$ (1.5 %, Domain 1) to 2.32 km$^2$ (9.5 %, Domain 7), while differences in Domain 8 amount to 1.04 km$^2$ (3.1 %). Variations in water depth between the highest and lowest configurations in Manning's $n$ coefficients are even smaller, varying between 1 cm (Domain 1, 8) and 2 cm (Domain 7) (2.1 % and 5.1 %, respectively). Due to the small differences in flood characteristics, we employed the moderate Manning's $n$ coefficients in our model setup (Table 2).

## 3.4 General Extreme Value Statistics (GEV)

To describe the distribution of ESL and return periods (RP), we employed the commonly used General Extreme Value (GEV) distribution (Coles, 2001). We use the time series of annual storm season maxima (July to June) of 30 tide gauges and 32 boundary stations (see Section 3.2.3). The GEV is defined by

$$F(t, \mu, \sigma, \xi) = \exp\left\{ -\left[ 1 + \xi\left(\frac{t - \mu}{\sigma}\right) \right]^{-1/\xi} \right\}, \tag{1}$$

where $t$ is the sea level, $\mu$ is the location parameter, $\sigma$ the scale parameter, and $\xi$ the shape parameter. For each gauge and boundary station used in this study, the GEV is fitted against the time series of the annual storm season´s maxima. In this study we use the Python code of Reinert et al. (2021) for the fitting. Using this model, we extrapolate 200-year return water levels for each tide gauge and boundary station. Note that slow, long-term variations in mean sea level have been subtracted from the time series by a linear fit. Therefore, we only consider ESLs relative to mean sea level in the statistics. Due to the microtidal regime of the Baltic Sea, the derived return periods and water levels correspond only to the surge component and neglect tidal contributions to ESL, as the latter (tide-surge interactions) are negligible in the study region (Arns et al., 2020).

## 3.5 Construction of the 200-year event

Since there are no tide gauge observations of storm surges with a return period of 200-years in the study region, we constructed the hydrographs (water level time series) for such events as input for the coastal inundation model: Within the modelled time frame from 1961 to 2018, we extracted for each boundary station all events exceeding a water level of one meter above mean sea level, which is the threshold for an ESL defined by the German Federal Maritime and Hydrographic Agency for the German Baltic Sea coast. We made sure that at least 48 hours separated the peak water levels of individual events. For the boundary stations where the water level never reached one meter, annual maxima were used instead, e.g. in the Saaler Bodden lagoon. Each extracted time series has a time step of one hour. Its water level is normalised by the maximum water level of the individual event and then interpolated to a time step of 15 min using cubic interpolation. The water level time series of each extracted surge starts three days before the peak water level and ends three days after, which covers the time period of an ESL event in the Baltic Sea of approximately 24 hours. This allows us to compute the mean evolution of a surge by taking the mean of all events per station (Figure 2a). By multiplying the mean, normalised evolution with the respective 200-year return level obtained from the GEV analysis, we have constructed the hydrographs of the 200-year event (see Figure 2b-f as examples). These hydrographs are then used as boundary conditions for the coastal inundation model.

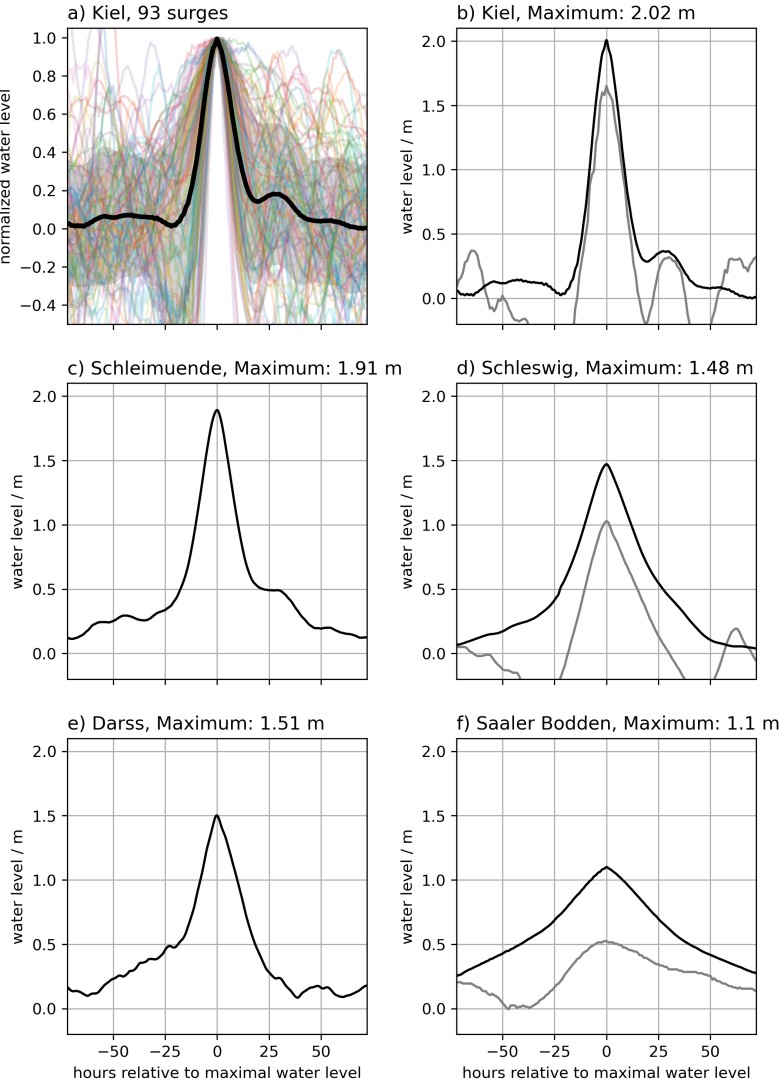

**Figure 2.** a) The normalised surge shape (thick black line), its standard deviation (grey area), and the individual time series (coloured) for the boundary station 'Kiel'. b) The constructed 200-year event for the boundary station 'Kiel'. c) The constructed 200-year event for the boundary station 'Schleimuende'. d) The constructed 200-year event for the boundary station 'Schleswig'. e) The constructed 200-year event for the boundary station 'Darss'. f) The constructed 200-year event for the boundary station 'Saaler Bodden'. The grey lines in b), d), and f) are the hydrographs of the 2019 surge from the respective boundary stations as a comparison.

### 3.6 Data used for model validation

#### 3.6.1 Tide gauge data to validate coastal ocean model

To evaluate the coastal ocean model's performance, we compared the modelled ESL to tide gauge observations listed in Tab. 3. In Section 4.1 we compare the model's capability to reproduce observed ESL (30 year return water level) and its ability to extrapolate 200-year return levels. We have used the full record length of each tide gauge, which is often longer than the hindcast period simulated with the coastal ocean model. Since only 'Gedser' (5) and 'Hornbaek' (8) are significantly longer, we have included the GEV return levels based on the overlapping time series for both tide gauges in Tab. 4. For each gauge, the SLR and other slow, long-term changes in the mean sea level have been removed by subtracting a linear fit of the time series.

#### 3.6.2 Geodetic levelling data to validate dike heights in coastal inundation model

We have used 9519 high accuracy real time kinematic (RTK) GPS points of the official dike crest geodetic levelling provided by the Schleswig-Holstein State Agency for Coastal Protection, National Park and Marine Conservation (LKN) to validate the dike heights extract from the DEM 1 (1 x 1 m horizontal resolution; Table 1). We compared the LKN RTK points and the dike crest elevations of the coastal inundation model within a 25 m buffer around the dikeline used in this study (Table 1). Next, in order to minimise errors of location (RTK points vs. 50 m x 50 m cells), we have removed all cells of the coastal inundation model that were outside the value range of the geodetic dike levelling.

#### 3.6.3 Sentinel-1 SAR imagery to validate flood extents simulated with coastal inundation model

We have used Sentinel-1 SAR imagery to compute the flood extent of the storm surge that occurred on January 2$^{nd}$ in the study area. The SAR image covers a large portion of the coastline of the federal state of SH and is therefore used for evaluating the output of the coastal inundation model. SAR satellite missions enable the monitoring of flood extent over large geographical areas and at high spatial resolution independent of cloud coverage and illumination conditions. The specular reflectance of radar pulses at the water surface results in low return signals at-sensor, which enables delineation of water bodies from other surfaces (Clement et al., 2018). The European Space Agency's Sentinel-1 (S-1) mission offers high spatial resolution C-band SAR imagery, whereby the operation of the twin satellites S-1A and B enables observations at an increased repeat cycle of six days, which is useful for capturing short-term events. S-1B acquired the region of interest in ascending orbit and Interferometric Wide (IW) swath instrument mode on January 02, 2019 at 17:08 UTC. We used Google Earth Engine to access the calibrated and ortho-corrected Ground Range Detected (GRD) product at 10 m spatial resolution (Google Earth Engine (GEE), 2022). After visual inspection, we used VV polarisation, and applied a focal median filter with a radius of 70 m to reduce noise. We then applied a simple threshold of -16 dB to compute a binary water map, and used the shorelines of lakes, supposedly unaffected by the surge, to visually assess its suitability.

**Table 3.** Overview of the tide gauges, their record lengths, and locations used in this study. We defined gaps in the time series as missing data with lengths greater than one day. The data is obtained from the European Marine Observation and Data Network (EMODnet, https://emodnet.ec.europa.eu).

| number | tide gauge name | record lengths | lon / lat | number of gaps |
|---|---|---|---|---|
| 0 | Althagen | 1953-11-01 to 2020-12-31 | 12.42 / 54.37 | 3 |
| 1 | Barhoeft | 1954-11-01 to 2020-12-31 | 13.03 / 54.43 | 1 |
| 2 | Barseback | 1982-04-26 to 2020-12-31 | 12.90 / 55.76 | None |
| 3 | Eckernfoerde | 1989-11-01 to 2020-12-31 | 9.84 / 54.47 | 3 |
| 4 | Flensburg | 1954-11-01 to 2020-12-25 | 9.43 / 54.79 | 4 |
| 5 | Gedser | 1891-09-01 to 2020-12-31 | 11.93 / 54.57 | None |
| 6 | Greifswald | 1963-11-01 to 2020-12-31 | 13.45 / 54.09 | None |
| 7 | Heiligenhafen | 1989-06-01 to 2020-12-31 | 11.01 / 54.37 | 6 |
| 8 | Hornbaek | 1891-01-01 to 2020-12-31 | 12.46 / 56.09 | None |
| 9 | Kappeln | 1991-11-01 to 2020-12-31 | 9.94 / 54.66 | None |
| 10 | KielHoltenau | 1964-11-01 to 2020-12-31 | 10.16 / 54.37 | None |
| 11 | Klagshamn | 1929-11-13 to 2020-12-31 | 12.89 / 55.52 | None |
| 12 | Koserow | 1972-11-01 to 2019-11-13 | 14.00 / 54.06 | 9 |
| 13 | LTKalkgrund | 1990-05-01 to 2020-12-31 | 9.89 / 54.82 | None |
| 14 | Langballigau | 1991-11-01 to 2020-12-31 | 9.65 / 54.82 | None |
| 15 | Neustadt | 1991-11-01 to 2020-12-31 | 10.81 / 54.10 | 1 |
| 16 | Rostock | 1968-11-01 to 2020-12-31 | 12.15 / 54.08 | 2 |
| 17 | Sassnitz | 1954-08-01 to 2020-12-31 | 13.64 / 54.51 | None |
| 18 | SchleimundeSP | 1990-11-01 to 2020-12-31 | 10.04 / 54.67 | None |
| 19 | Schleswig | 1991-11-01 to 2020-12-31 | 9.57 / 54.51 | None |
| 20 | Simrishamn | 1982-05-31 to 2020-12-31 | 14.36 / 55.56 | None |
| 21 | Skanor | 1992-02-17 to 2020-12-31 | 12.83 / 55.42 | None |
| 22 | Stralsund | 1961-11-01 to 2020-12-31 | 13.10 / 54.32 | 2 |
| 23 | Timmendorf | 1961-11-01 to 2020-12-31 | 11.38 / 53.99 | 1 |
| 24 | Travemunde | 1949-11-01 to 2020-12-31 | 10.87 / 53.95 | 1 |
| 25 | Ueckermuende | 1965-11-01 to 2020-12-31 | 14.07 / 53.75 | 3 |
| 26 | Viken | 1976-04-22 to 2020-12-31 | 12.58 / 56.14 | None |
| 27 | Warnemuende | 1953-11-01 to 2020-12-31 | 12.10 / 54.17 | 5 |
| 28 | Wismar | 1957-11-01 to 2020-12-31 | 11.46 / 53.90 | 5 |
| 29 | Wolgast | 1965-11-01 to 2020-12-31 | 13.77 / 54.04 | 2 |

## 4 Results and discussion

### 4.1 Validation of extrapolated extreme sea levels

From the hindcast simulation between 1961 and 2018, we first compared the observed and modelled ESL events with water levels exceeding one meter above mean sea level (Fig. 3). For each tide gauge (Tab. 3), we computed the bias and standard deviation between observed and modelled ESLs (see Fig. 3a for the tide gauge 'Kiel Holtenau' (10) as an example). Overall, the model has a negative bias, underestimating the ESLs by 11 cm on average (Fig. 3b).

Despite the negative bias of the ESLs, the model can reproduce the GEV distributions for most tide gauges (see Fig. 4 for 'Kiel Holtenau' (10) and Tab. 4 where the return levels for a 30-year event and 200-year event are listed). For the tide gauges with the longest time series, 'Gedser' (5) and 'Hornbaek' (8), the model shows higher return levels than the observations when the observed time series are clipped to the model time period before analysis. The reason for this is that the largest surge at those locations lies outside the modelled time period, which changes the tail of the GEV distributions. This is a common problem in extreme value statistics. An event with a high return level can change the tail of the distribution and thus the extrapolation. Similarly, from the negative bias of the model for the high surges (Fig. 3a) one would expect that also the GEV statistics show a clear negative bias for the high return periods. However, one overestimated event (the highest surge in the hindcast period) is enough to 'fix' the tail of the GEV distribution in this case. Nevertheless, the 95% confidence intervals from the model statistics include the return levels of the GEV of the clipped time series. For 'Althagen' (0), 'Kappeln' (9), and 'Schleswig' (19), the model overestimates the return levels since the lagoons are not sufficiently resolved at the resolution of 200 m. Because of the deviations at these locations, we used the tide gauge observations as input for the coastal inundation model instead.

For 'Barseback' (2) and 'Viken' (26) the model underestimates the return levels, as also shown in Fig. 3b, with 'Barseback' (2) having the largest negative bias (note that both tide gauges are located in Sweden). The spatial distribution of the 30-year and 200-year return levels (Fig. 5) shows that the highest ESLs occur at the coast, with decreasing ESLs from west to east. This pattern has already been described in the literature (e.g. Gräwe and Burchard, 2012; Wolski et al., 2014), and is primarily due to fetch length, which is longer for the coast of SH compared to MP during (north) easterly winds.

### 4.2 Validation of dike height extraction from DEM 1

Knowledge on the elevation of dike heights constitutes a major challenge for large scale flood risk simulations as data on flood protection standards are scarce and impacts have been shown to be most sensitive to variations in adaptation strategies (Hinkel et al., 2014; Vousdoukas et al., 2016; Scussolini et al., 2016; Hinkel et al., 2021). Consequently, the validation of dike heights in DEMs used to simulate coastal flooding is crucial for assessing the validity of modelled flood extents.

Here we show that the use of high resolution (1 x 1 m) DEMs for extracting dike heights can still lead to deviations between modelled (DEM) and measured (RTK) dike crest elevations. However, we attribute the great majority of these deviations to issues related to the positioning of the raster cells, which were extracted within a 25 m buffer around the dikeline (see Section 3.6.2). The comparison of 50 m x 50 m cells around curvy dikelines with RTK points can lead to positional errors, where cells overlaying the RTK points may represent neighbouring hills or troughs rather than the actual dikeline. This is also supported

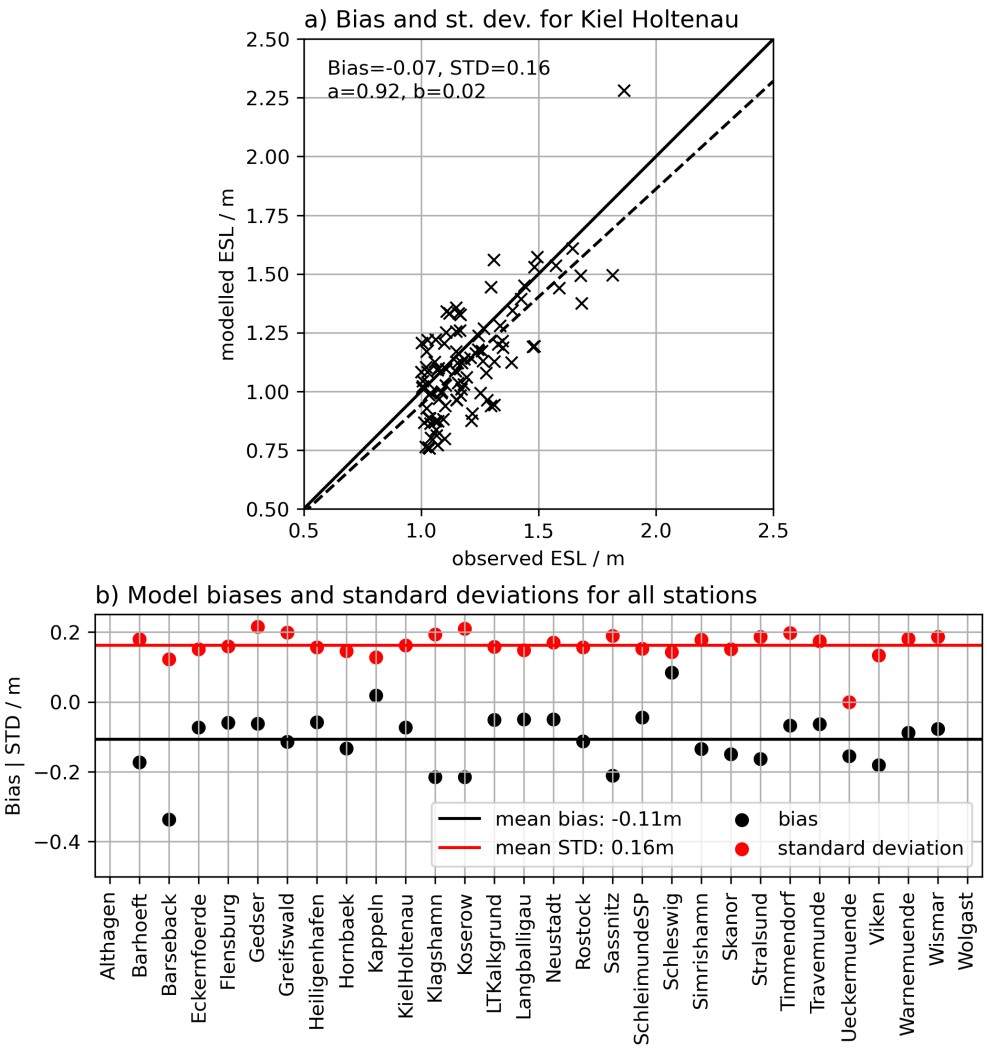

**Figure 3.** Validation of the hydrodynamic model's extreme sea levels exceeding 1 meter height above mean sea level. a) Comparison of the ESLs for the tide gauge 'Kiel Holtenau' (10) in terms of bias and standard deviation (STD). The dashed line shows a linear fit where the slope denotes if the model's bias is depending on the height of the ESL. b) Summary of the biases and standard deviations for all stations. The mean bias along all stations is −11 cm, indicating that in general the ESLs are underestimated by the model. Note that no water levels above one meter are observed for the tide gauges 'Althagen' (0) and 'Wolgast' (29).

**Table 4.** List of the return levels (RL) of the 30-year and 200-year return periods including the 95 % confidence intervals for different tide gauges along the Baltic Sea coast. RL were estimated from observations (obs.) and model results (mod.).

| number | tide gauge name | obs. 30-year RL / m | mod. 30-year RL / m | obs. 200-year RL / m | mod. 200-year RL / m |
|---|---|---|---|---|---|
| 0 | Althagen | $0.92 \pm 0.24$ | $1.36 \pm 0.19$ | $1.10 \pm 0.36$ | $1.71 \pm 0.44$ |
| 1 | Barhoeft | $1.35 \pm 0.08$ | $1.33 \pm 0.10$ | $1.44 \pm 0.12$ | $1.53 \pm 0.12$ |
| 2 | Barseback | $1.31 \pm 0.13$ | $0.91 \pm 0.08$ | $1.45 \pm 0.19$ | $1.00 \pm 0.16$ |
| 3 | Eckernfoerde | $1.69 \pm 0.21$ | $1.74 \pm 0.16$ | $1.91 \pm 0.39$ | $2.05 \pm 0.28$ |
| 4 | Flensburg | $1.63 \pm 0.11$ | $1.72 \pm 0.16$ | $1.80 \pm 0.21$ | $2.03 \pm 0.29$ |
| 5 | Gedser | $1.51 \pm 0.08$ | $1.51 \pm 0.13$ | $1.71 \pm 0.11$ | $1.74 \pm 0.26$ |
| 5 | Gedser (overlap) | $1.39 \pm 0.09$ | $1.51 \pm 0.13$ | $1.51 \pm 0.15$ | $1.74 \pm 0.26$ |
| 6 | Greifswald | $1.49 \pm 0.12$ | $1.49 \pm 0.12$ | $1.65 \pm 0.22$ | $1.68 \pm 0.23$ |
| 7 | Heiligenhafen | $1.72 \pm 0.35$ | $1.61 \pm 0.17$ | $2.15 \pm 0.83$ | $1.90 \pm 0.35$ |
| 8 | Hornbaek | $1.49 \pm 0.06$ | $1.43 \pm 0.13$ | $1.64 \pm 0.07$ | $1.60 \pm 0.25$ |
| 8 | Hornbaek (overlap) | $1.47 \pm 0.06$ | $1.43 \pm 0.13$ | $1.56 \pm 0.07$ | $1.60 \pm 0.25$ |
| 9 | Kappeln | $1.36 \pm 0.13$ | $1.66 \pm 0.18$ | $1.51 \pm 0.23$ | $2.01 \pm 0.34$ |
| 10 | KielHoltenau | $1.73 \pm 0.18$ | $1.73 \pm 0.16$ | $1.99 \pm 0.36$ | $2.03 \pm 0.29$ |
| 11 | Klagshamn | $1.13 \pm 0.08$ | $1.20 \pm 0.11$ | $1.27 \pm 0.16$ | $1.37 \pm 0.21$ |
| 12 | Koserow | $1.42 \pm 0.05$ | $1.38 \pm 0.11$ | $1.47 \pm 0.07$ | $1.56 \pm 0.26$ |
| 13 | Langballigau | $1.66 \pm 0.26$ | $1.69 \pm 0.16$ | $1.93 \pm 0.56$ | $1.99 \pm 0.28$ |
| 14 | LTKalkgrund | $1.67 \pm 0.29$ | $1.67 \pm 0.15$ | $2.03 \pm 0.65$ | $1.96 \pm 0.27$ |
| 15 | Neustadt | $1.64 \pm 0.22$ | $1.69 \pm 0.17$ | $1.85 \pm 0.43$ | $1.99 \pm 0.33$ |
| 16 | Rostock | $1.55 \pm 0.19$ | $1.52 \pm 0.15$ | $1.83 \pm 0.39$ | $1.75 \pm 0.31$ |
| 17 | Sassnitz | $1.22 \pm 0.08$ | $1.24 \pm 0.08$ | $1.33 \pm 0.14$ | $1.44 \pm 0.13$ |
| 18 | SchleimundeSP | $1.62 \pm 0.24$ | $1.69 \pm 0.15$ | $1.88 \pm 0.49$ | $1.98 \pm 0.26$ |
| 19 | Schleswig | $1.39 \pm 0.11$ | $1.91 \pm 0.28$ | $1.48 \pm 0.21$ | $2.44 \pm 0.65$ |
| 20 | Simrishamn | $1.09 \pm 0.09$ | $1.19 \pm 0.11$ | $1.19 \pm 0.15$ | $1.37 \pm 0.23$ |
| 21 | Skanor | $1.27 \pm 0.21$ | $1.31 \pm 0.10$ | $1.46 \pm 0.44$ | $1.48 \pm 0.19$ |
| 22 | Stralsund | $1.41 \pm 0.13$ | $1.39 \pm 0.11$ | $1.59 \pm 0.23$ | $1.58 \pm 0.21$ |
| 23 | Timmendorf | $1.65 \pm 0.18$ | $1.65 \pm 0.18$ | $1.93 \pm 0.38$ | $1.94 \pm 0.38$ |
| 24 | Travemunde | $1.71 \pm 0.13$ | $1.72 \pm 0.18$ | $1.92 \pm 0.24$ | $2.03 \pm 0.36$ |
| 25 | Ueckermuende | $1.00 \pm 0.12$ | $0.90 \pm 0.13$ | $1.15 \pm 0.25$ | $1.11 \pm 0.30$ |
| 26 | Viken | $1.49 \pm 0.16$ | $1.32 \pm 0.13$ | $1.69 \pm 0.31$ | $1.49 \pm 0.25$ |
| 27 | Warnemuende | $1.54 \pm 0.16$ | $1.50 \pm 0.15$ | $1.81 \pm 0.35$ | $1.74 \pm 0.32$ |
| 28 | Wismar | $1.69 \pm 0.13$ | $1.68 \pm 0.19$ | $1.89 \pm 0.23$ | $1.97 \pm 0.40$ |
| 29 | Wolgast | $0.96 \pm 0.11$ | $0.95 \pm 0.15$ | $1.09 \pm 0.23$ | $1.21 \pm 0.40$ |

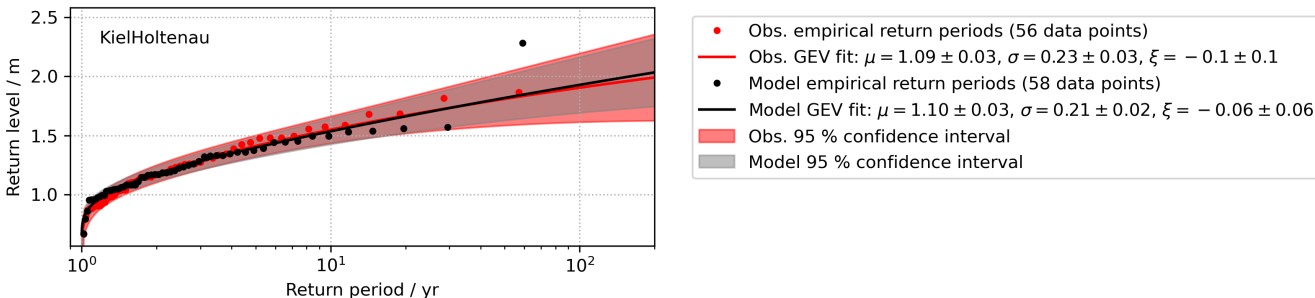

**Figure 4.** GEV distributions derived from the observations (red) and the coastal ocean model (black) for the station 'Kiel Holtenau' (10).

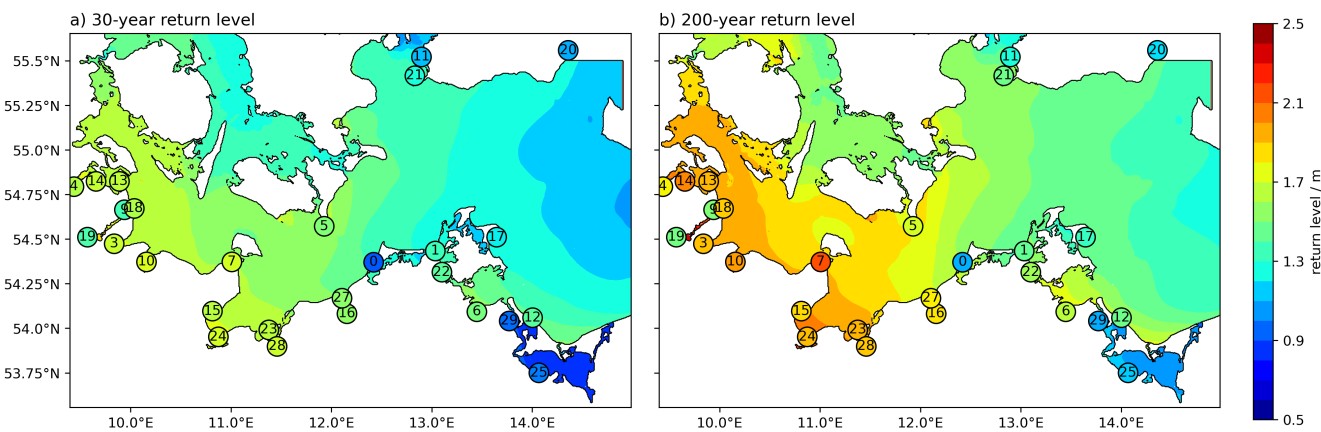

**Figure 5.** Modelled 30-year (a) and 200-year return levels (b) for the western Baltic Sea. The circles denote the observed tide gauge locations and the values listed in Tab. 4. Note that the values are overestimated for sheltered lagoons and semi-enclosed water bodies with narrow inlets such as The Schlei (located between the Baltic Sea and the city of Schleswig, see Fig. 1, 7), and the 'Saaler Bodden' ('Althagen', 0), because of the limited resolution (Tab. 4). Note that some of the tide gauges listed in Tab. 3 are located north of the domain shown.

by the fact that the lowest dike heights in the coastal inundation model tend to produce higher differences when compared to the RTK measurements (and vice versa) (Appendix Fig. A2).

Given the differences in scale (points vs. 50 m x 50 m cells), we find that the RTK measurements and the dike heights used in the coastal inundation model are generally in good agreement (Appendix Fig. A3). The minimum (-2 m) and maximum (3.7 m) deviations can be substantial, while the root-mean-square-error between both datasets is 0.65 m and the mean absolute error 0.37 m. However, 62 % of the values lie in the range between + 0.2 m and - 0.2 m difference and 77 % between - 0.5 m and + 0.5 m difference.

 **4.3 Validation of maximum sea levels during the 2nd January 2019 event**

Comparing the peak water levels of the coastal ocean model with the observed maxima of the 2019 event (Fig. 6) shows that the model can capture the spatial pattern of water level variability. Overall, the model underestimates peak water levels by 5 cm on average (more for the higher water levels), yet overestimates the water levels in most lagoons, as already stated in Section 4.1. The root-mean-square-error between modelled and measured peak water levels is 15 cm.

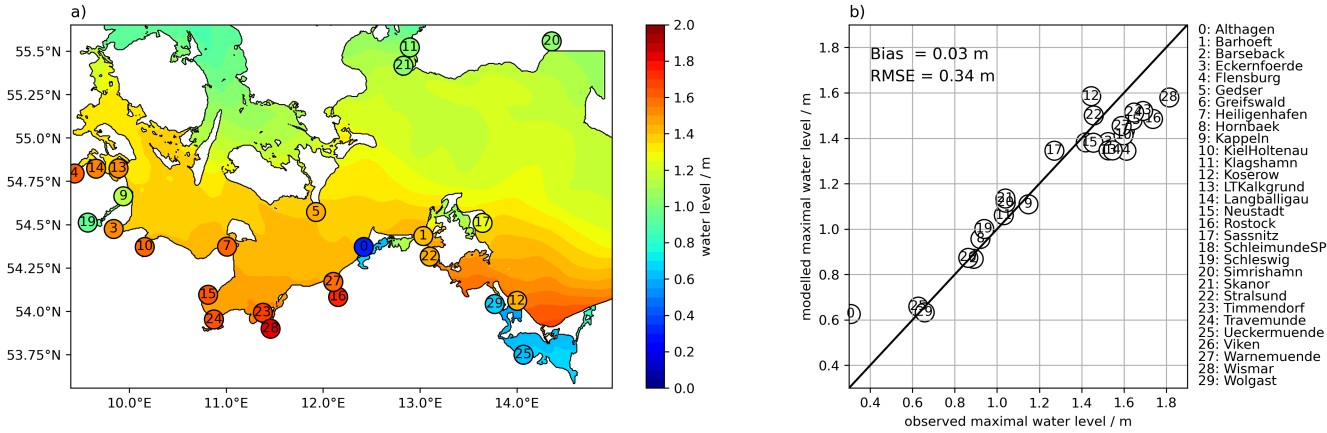

**Figure 6.** a) Modelled and observed maximum water levels (coloured circles) for the storm event of January 2nd 2019 across the study region. b) Direct comparison of the observed maximum water levels with the modelled maximum water levels. The mean bias of the model to the observed stations is −5 cm, i.e. on average the model underestimates the maximum water levels. Note that some of the stations listed in Tab. 3 are north of the domain shown.

**4.4 Comparison of simulated coastal flood extents with SAR derived imagery**

We used the flood extent derived from S-1 SAR imagery (acquired at 17:08 UTC on the 2nd January 2019) to evaluate the validity of the flood extents simulated with the coastal inundation model. The suitability of SAR imagery for validating coastal flood extents has previously been described in Eilander et al. (2023). SAR imagery was acquired 1.5 to 3 hours after the modelled peak of the surge in Schleimünde (18) (entrance of the ca. 41 km long subglacial meltwater channel called The Schlei, which is located between the Baltic Sea and the city of Schleswig) and Timmendorf (23) (north of Lübeck), and 3.5 hours after water levels were peaking in Kiel fjord. Only at the end of The Schlei, in Schleswig (19), the modelled peak of the surge occurred at 23:15, more than six hours after the SAR imagery was taken. When the SAR image was acquired, water level at Schleswig (19) was at 0.64 m, which is 0.35 m below the peak water level of the 2019 event at the same location (0.99 m) (see also Fig. A4 of the appendix for water level time series of the 2019 event for selected boundary stations).

The comparison of flood extents derived from SAR imagery and hydrodynamic models is compromised by several issues. Mapping surface water from SAR imagery is based on contrast, resulting from low return signals due to specular reflectance of radar pulses at the water surface. Consequently, problems may arise if this assumption is compromised. This may be the

case if other surface types result in similar weak backscatter signals (e.g., smooth tarmac, dry sandy soils, or wet snow), which may result in overestimating surface water areas. On the contrary, relatively high backscatter of surface water areas (e.g.,

caused by emerged vegetation or roughening of the water surface due to wind or rain) reduces the contrast between water and adjacent terrestrial surfaces, typically resulting in an underestimation of surface water area (Chini et al., 2021). In another comparison, Mason et al. (2009) found that the similar signal of un-flooded short vegetation and adjacent flood plains was a main error in waterline positioning. Strong winds accompanying the 2019 event may likely have caused a softening of the contrast between open water and other surfaces with similar backscattering characteristics, which might explain some of the

differences. Other than this, the closing of flood gates to prevent seaside flooding may have resulted in accumulation of riverine water and corresponding inundation inland, while the opening of flood gates to promote inundation of certain areas during surges (e.g., for reasons of nature conservation) may have resulted in flooding of areas protected by hard defenses, and thus would not have been considered by the hydrodynamic model.

In addition, the pre-processing of the elevation data (within a 100 m buffer around the coastline, each 50 m grid cell is given

the maximum elevation identified in the 1 m x 1 m DEM) results in a general overestimation of nearshore elevations and 50 m resolution is often too coarse to accurately resolve many narrow beaches in the study region. We must note, therefore, that we do not consider the comparison of SAR-derived inundation maps and the simulated coastal flood extents of the coastal inundation model as validation in the strict sense of the term but rather as a first-order evaluation of the model's performance.

Without accounting for the uncertainties mentioned above when validating flood extents on large spatial scales, as was done for this study, the resulting skill indices can leave a misleading impression on model performance. Therefore, we tried to account for these uncertainties by adjusting the SAR-derived and modelled flood extents prior to comparison. The flooded area was calculated as follows: first, we excluded all flooded areas inside the 100 m coastline buffer in the SAR-derived and coastal inundation model floodplains. In addition, beach lakes and lagoons cut off from the Baltic Sea by sluice gates or beach

ridges (indicated by ATKIS® digital landscape model as stagnant water) were also excluded. We compared the flooded area of both datasets (SAR and coastal inundation model) by calculating the percentage of correctly predicted flood extent (agreement in inundated area between SAR-imagery and coastal inundation model), missed flood extent (inundation observed in the SAR-imagery but not in the coastal inundation model), and overpredicted flood extent (inundation simulated with the coastal inundation model but not observed in SAR-imagery) (Fig. 7). These indices are based on the indices commonly used in the cur-

rent literature to estimate the validity of the output of hydrodynamic models (Vousdoukas et al., 2016; Alfieri et al., 2014). For the area covered by the SAR image (Fig. 7), the storm surge from 2[nd] January 2019 produced a (corrected) flood extent of 2.18 km$^2$ and 2.38 km$^2$ for the SAR imagery and the coastal inundation model, respectively. Relative to the SAR data, we calculated that 50 % of the inundated area was correctly predicted, 50 % was overpredicted, and 50 % of the flooded area was missed.

Problematically, the correction of the flood extent prior to comparison with the SAR-imagery within a 100 m buffer around the coastline leads to a substantial reduction of the flooded area along peninsulas, such as the flood prone sand spits of the study region (Fig. 7b vs. Fig. 7c). This, again, negatively biases the skill indices presented above.

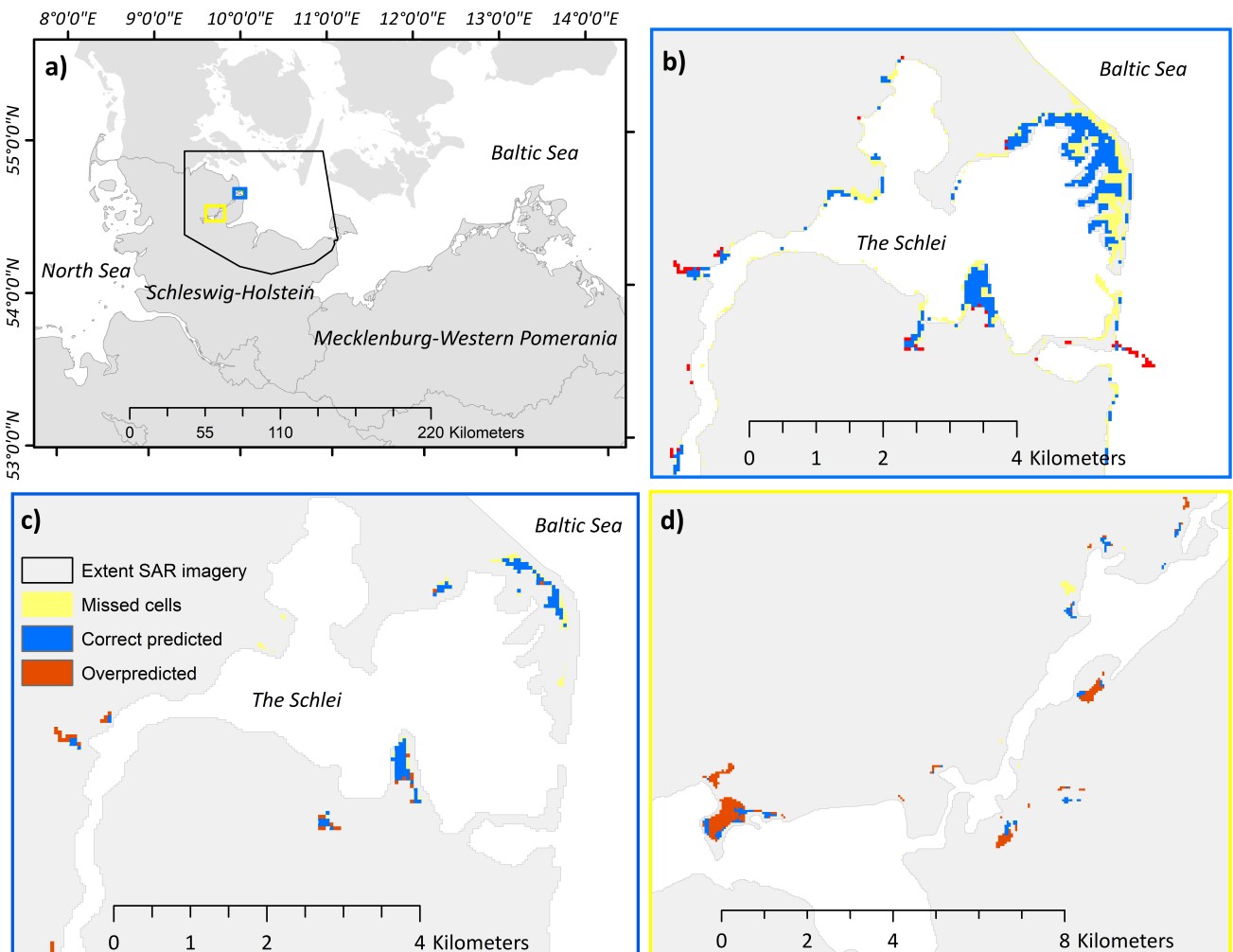

**Figure 7.** Example regions showing the comparison between flood extents produced with the coastal inundation model and extracted from the SAR-imagery for the storm surge of January 2[nd] 2019. a) Overview map showing locations of example regions and spatial coverage of SAR-imagery. b) Comparison of flood extents prior to any of the corrections described above. c) Comparison of flood extents after correction for the same region as depicted in b). d) Comparison of flood extents for another section of The Schlei, near the city of Schleswig. The skill indices provided above are based on the corrected flood extents for the whole coverage of the SAR-imagery (a)), as shown in c) and d).

In the absence of measured *in situ* data on coastal inundation characteristics during storm surges, we propose the use of remote sensing products as a promising alternative for the validation of hydrodynamic models. Yet, the validation of SAR derived flood maps is challenging. The use of remote sensing products for flood extent validation may be better suited for smaller, connected flood extents, where the above limitations can be more easily addressed. The strong potential for such applications has previously been shown by Eilander et al. (2023); Vousdoukas et al. (2016). We show that the focus on specific

areas with connected flood extents would also result in a higher agreement between the SAR-imagery and our simulation (Fig. 7b).

A more general challenge for the comparison of satellite derived flood extents with hydrodynamic model results is the generally short duration of ESL. Even if the latter is already quite long-lasting along the Baltic Sea coast compared to macrotidal environments (Wolski and Wiśniewski, 2020; MacPherson et al., 2019), the limited duration reduces the chance of suitable matches with satellite observations. Nevertheless, the growing number of satellite missions may improve the availability of relevant observations in the future.

We believe that incorporating satellite-derived flood extents in the portfolio of potential validation data may increase the opportunities to validate hydrodynamic flood models. However, such products are still dependent on algorithm development and limited by spatio-temporal coverage.

## 4.5    Flood characteristics of simulated storm surge scenarios

Our results confirm that the German Baltic Sea coast is exposed to coastal flooding, as a storm surge with a return period
of 200-years already produces substantial inundated areas (Fig. 8, Table 5). Without upgrading existing coastal protection infrastructure until 2100, the flood extent can increase by almost a factor of five when SLR reaches 1.5 m. For the four storm surge scenarios, the majority of the flooded area is located in the federal state of MP, varying between 85 % and 89 %, whereas the contribution of SH varies between 11 % and 15 % (Table 5).

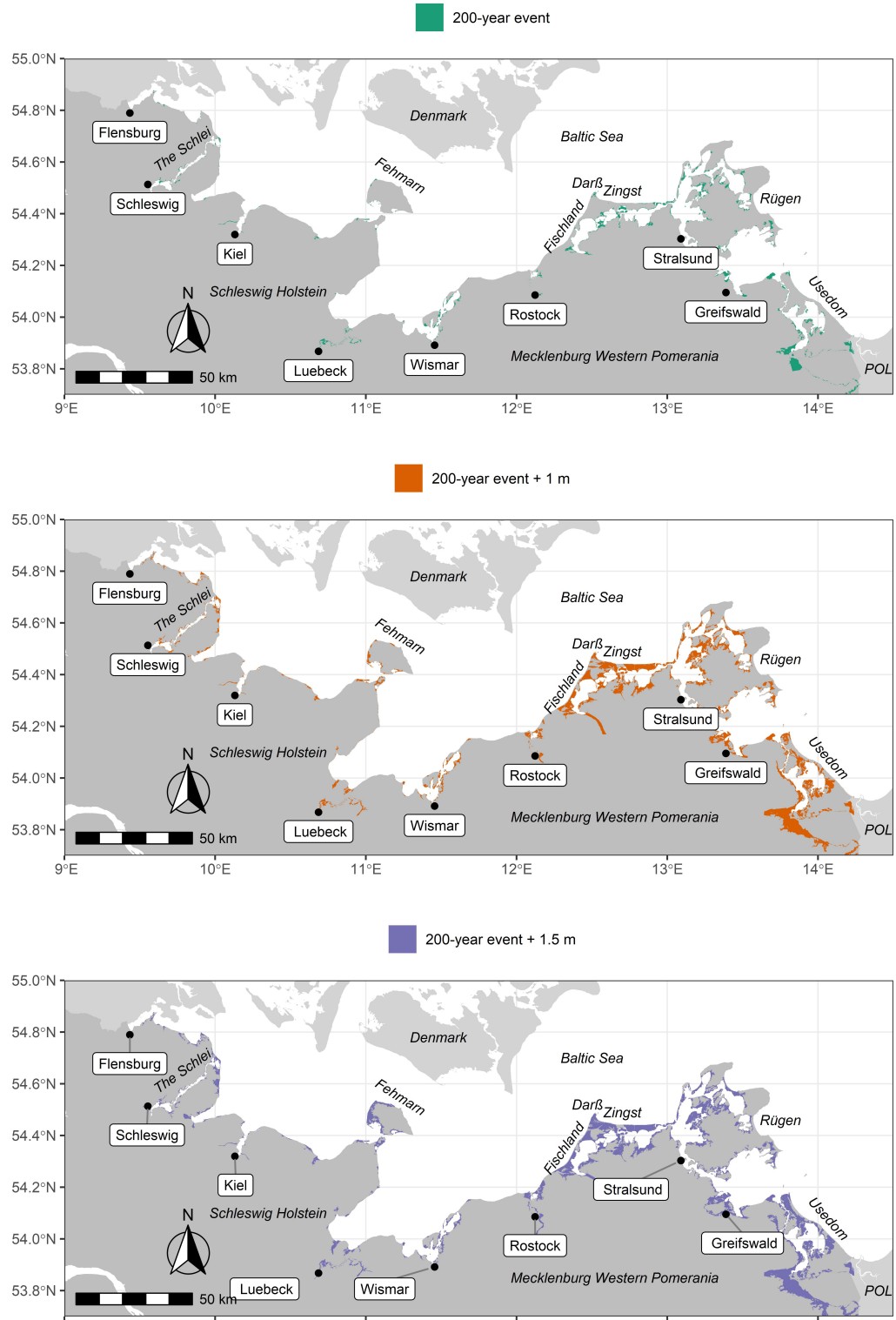

**Figure 8.** Flood extent for a storm surge with a return period of 200 years (top), the same surge and 1 m (middle), and 1.5 m SLR (bottom).

**Table 5.** Flood characteristics for the four storm surge scenarios for MP, SH and the entire German Baltic Sea coast.

|  |  | MP | SH | total |
|---|---|---|---|---|
| Storm surge 2019 | flood extent km$^2$ | 101.26 | 16.44 | 117.61 |
|  | average max. flood depth (m) | 0.59 | 0.9 | 0.63 |
| 200-year event | flood extent km$^2$ | 191.54 | 25.31 | 216.84 |
|  | average max. flood depth (m) | 0.63 | 1.03 | 0.67 |
| 200-year event + 1 m | flood extent km$^2$ | 673.33 | 79.77 | 753.09 |
|  | average max. flood depth (m) | 1.31 | 1.25 | 1.31 |
| 200-year event + 1.5 m | flood extent km$^2$ | 868.48 | 147.62 | 1016.10 |
|  | average max. flood depth (m) | 1.64 | 1.5 | 1.62 |

In contrast to the flood extent, the average maximum inundation depths for the 2019 surge and the 200-year event are considerably higher in SH compared to MP (Table 5). The largest differences are found for the 200-year event, where water depths are 40 cm higher in SH. The differences converge under the influence of SLR. For both SLR scenarios, inundation depths are higher in MP. The analysis of flood depths during storm surges can be crucial, as it constitutes a major driver of potential flood damages to buildings and infrastructure (Merz et al., 2010; de Moel and Aerts, 2011).

Following Lopes et al. (2022), we show that the flood depth and extent along the German Baltic Sea coast are highly dependent on ESL and local geomorphological features, which can also explain the observed differences between the federal states of SH and MP. For instance, the 200-year event is characterised by peak water levels that are on average 0.46 m lower in MP compared to SH (2.03 m in SH and 1.57 m in MP, Table 4). In both SLR-scenarios, the flood depth is slightly higher in MP, which we attribute to the substantially larger flood extent and lower elevations within the floodplain. Within the flood extent of the 200-year event and 1.5 m of SLR, the mean elevation is 1.1 m NHN in MP compared to 1.27 m NHN in SH.

The importance of geomorphology for the inundated areas in our study region becomes also evident when comparing flood extents between the federal states of SH and MP (Table 5). Despite of lower peak water levels during the 2019 and the 200-year event without SLR, MP contains much larger flood extents, which is not exclusively due to the fact that MP comprises a substantially longer coastline and thus, the potential for larger floodplains. In SH, the flood extent per km of coastline varies between 0.025 km$^2$ km$^{-1}$ for the 2019 surge and 0.23 km$^2$ km$^{-1}$ for the 200-year event and 1.5 m of SLR. In contrast, the coastal length normalised flood extent in MP varies between 0.05 km$^2$ km$^{-1}$ for the 2019 event and 0.46 km$^2$ km$^{-1}$ for the highest surge scenario.

We identify several hotspots of coastal flooding along the German Baltic Sea coast. Overall, the majority of the flood extent is located along sheltered lagoons and estuaries. In SH, hotspots are found along the Flensburg fjord, The Schlei, Eckernförde (domain 1, counted from left to right in Fig. 1), Fehmarn (domain 3) and along the Trave estuary until the city of Lübeck (domain 4). In MP, the largest flood extents are identified in Rostock-Warnemünde (domain 6), Fischland-Darß-Zingst (domain 7), western and central Rügen (domains 8 and 9), north of Greifswald (domain 10), the island of Usedom and the Peene estuary,

located west of the island of Usedom (domain 11). Across both federal states, the largest coherent flood extents are observed in MP and comprise the lagoons of Fischland-Darß-Zingst, Rügen and Usedom (Fig. 8).

## 4.6 Assessing the validity of the model results

We find that the coastal ocean model, which provides the boundary conditions for the coastal inundation model, overestimates both the extrapolated ESL (the modelled 200-year return water level) and the peak water levels of the 2019 surge inside protected lagoons of Fischland-Darß-Zingst in MP and The Schlei in SH (Fig. 1, 6, Tab. 4). We have therefore used ESL extrapolations from the tide gauges Althagen (0), Kappeln (9) and Schleswig (19) as input for the coastal inundation model. We note that this change is only valid for this specific application and should not be considered part of the modelling framework.

Thus, the provision of boundary conditions from coarser resolution hydrodynamic models for sheltered coastal environments constitutes an extra source of uncertainty in large-scale coastal flood risk assessments that should be accounted for in future applications.

Only a few studies have examined the impact of storm surge duration and intensity on flood characteristics, but Höffken et al. (2020) have shown for a case study in the German Baltic Sea that flood extents can vary by 20 % when sea levels rise. The

450 Baltic Sea is characterised by a microtidal regime, which means that high water levels during storm surges can stay for several days and various storm surge intensities are observed (MacPherson et al., 2019). Consequently, storm surge hydrographs are spatially and temporally (between different storm surge events) variable (compare also Fig. 2a). While we account for the spatial variability by calculating mean storm surge hydrographs for 32 flood boundary stations (Fig. 1) across the study region, the temporal variability is not accounted for as we apply mean surge shapes. We note, therefore, that the flood extents shown

in this study can both be larger or smaller depending on the intensity and duration of the surge.

We consider the flood maps presented here as conservative for several reasons. First, the exclusion of waves in coastal flood modelling can lead to underestimations of flood depth and extent. A long series of breaking waves can substantially increase peak water levels due to wave setup and swash (also referred to as wave run-up) (Weisse et al., 2021; Melet et al., 2018). For

example, in a study from the Gulf of Finland, the contribution of wave setup to extreme water levels approached up to 50 %, while the maximum absolute contribution to peak water levels varied between 70 cm and 80 cm in exposed areas (Soomere et al., 2013). Although these values should be lower in the German Baltic Sea, where recorded maximum significant wave heights are considerably lower than in the Gulf of Finland (Alari, 2013), waves can affect coastal flood extents. This has also been shown on a pan-European scale (Vousdoukas et al., 2016). We still excluded wave setup in our analysis. A technical

reason is that the coastal ocean model has a resolution of 200 m and thus, cannot resolve the near shore (wave breaking zone) sufficiently to reproduce wave setup. In addition, we calibrated the coastal ocean model by increasing the wind speed by 7%. This allowed us to minimize the error in predicting the peak water levels and, using this method, we consider missing processes such as wave formation, lack of resolution and errors in the atmospheric model. Yet, the reason for the underestimation of high events is not entirely clear. Despite the 7% increase of wind speed, the underestimation may be partially explained by the

missing process of wave setup. Another reason could be that those few storms that were responsible for the underestimated

surges are not reproduced well by the atmospheric data (Lorenz and Gräwe, 2023). Nevertheless, the model GEV distribution overestimates the 200-year return levels compared to the tide gauges. Thus, we expect the uncertainty of wave setup on the flood maps to be small, especially for the 200-year cases with and without SLR. In addition, there is still no conclusive information on potential future changes in the wave climate, and existing results show strong spatio-temporal variability (Weisse et al., 2021).

Another reason why we believe our model results can be considered conservative is that we overestimate dike heights in the coastal inundation model. The representation of subgrid-scale coastal adaptation measures such as dikes in coastal flood modelling constitutes a major challenge that can affect simulated flood extents (Hinkel et al., 2014, 2021; Vousdoukas et al., 2016). We have shown that dike crest elevations used in the coastal inundation model are generally in good agreement with high accuracy RTK measurements (see Section 4.2). However, there are cells that deviate substantially from the overlying RTK points, challenging the validity of the flood extents presented here. In order to test the sensitivity of the coastal inundation model to variations in dike crest elevations, we set up the coastal inundation model using the average RTK dike height for each pixel overlaying the dike line. Running the adjusted model for the 200-year event produced a flood extent of 27.8 km$^2$, which is 9.8 % higher than in the original setup (compare Table 5). Therefore, we believe that the dike crest elevations used in the coastal inundation model are reliable. However, we must stress that in this study, we had access to high resolution elevation data, which is very rare in large-scale flood modelling, suggesting that the sensitivity of most flood models to variations in dike crest elevations or elevation models in general is likely to be higher.

Finally, the results presented here do not account for morphological responses to rising water levels, such as the potential of shoreline change, dune collapse and dike breaching. Dike failure can occur due to hydraulic loads induced by waves and water level (Marijnissen et al., 2021), and thus, flooding behind embankments may not exclusively occur due to wave overtopping or overflow. Large-scale dike overflow is observed particularly for the 200-year event and 1.5 m of SLR, as the dike heights in the coastal inundation model represent the status quo (i.e. without potential future increases in dike height). However, the ongoing implementation of so-called climate dikes in the study region allows the dike elevation to be increased by up to 1.5 m (Melund, 2022). Further adjustments even allow the increase of dike heights by up to 2 m (Hofstede and Hamann, 2022). Ignoring the potential of dike failure, we therefore expect that the increase in dike heights by 1.5 m could offset a fraction of the additional flood extent caused by the SLR scenario. On the other hand, the increase in dike heights will be very costly (although probably cheaper than the expected flood damage costs (Hinkel et al., 2014)) and may not be applicable to many regional dikes that are characterised by variable safety standards (Melund, 2022). Therefore, and in the light of the findings presented here, we agree that the development and identification of new and complementary measures to mitigate increasing coastal flood risks constitutes one of the most prominent challenges facing coastal communities today (Morris et al., 2018).

## 5 Conclusions

In this study, we show that the current design heights of dikes along the German Baltic Sea coast are not sufficient to prevent flooding during a storm surge with a 200-year return period under high sea-level rise scenarios. Hotspots of coastal flooding are

mainly located in the federal state of Mecklenburg-Western Pomerania, where the lagoons of Fischland-Darß-Zingst, Rügen and south of Usedom (Szczecin Lagoon) are particularly exposed (Fig. 8). With foresight, the state authorities in Schleswig-Holstein and Mecklenburg-Western Pomerania have initiated the upgrade of state dike heights until the end of the century, which will allow comparatively easy increases of up to 1.5 m (Melund, 2022; Hofstede and Hamann, 2022; StALU, 2012). However, the effectiveness of dike height increases to compensate for high SLR scenarios has not yet been demonstrated. In addition, the associated costs as well as the future of many regional dikes with variable design heights (Melund, 2022) remain uncertain. Some regional dikes may become the responsibility of the federal states, but their maintenance and the increase in height required to compensate for high SLR scenarios (such as 1.5 m) may foster the need to rethink contemporary coastal protection measures towards new, more nature-based solutions.

In line with previous studies, we find that model validation (of all model components used) remains one of the biggest problems in large-scale flood modelling. With respect to the validation of flood extents, we show that besides the often used vertical aerial photography, other remote sensing products such as SAR-imagery may provide a promising alternative. Currently available spatio-temporal resolutions and problems associated with the detection of surge-driven coastal flooding versus wet soils, e.g. as a consequence of rainfall, compromise current applications, particularly on large spatial scales as presented in this study. The growing number of satellite missions may improve the availability of suitable observations in the future.

We suggest future research in the region to improve our understanding on 1) potential future changes in wave climate and associated impacts on coastal flood extents; 2) morphodynamic responses of natural and anthropogenic flood barriers to high water levels and wave loading with a particular focus on dike breaching and; 3) water level dynamics and vulnerability of low lying sheltered lagoons and inlets to storm surges and SLR.

*Data availability.* The flood boundary stations, associated water level timeseries representing the four storm surge scenarios (2019 surge, 200-year event and the 200-year event plus 1 m and 1.5 m of SLR), the simulated flood characteristics (flood extent and depth), the spatially explicit results of the extreme value analysis for every grid cell of the coastal ocean model, the modelled monthly peak water levels between 1961 and 2018 for every grid cell of the coastal ocean model and the modelled timeseries of water levels during the 2019 storm surge and for the entire hindcast period are freely available from Kiesel et al. (2023).

# 1   Appendix A

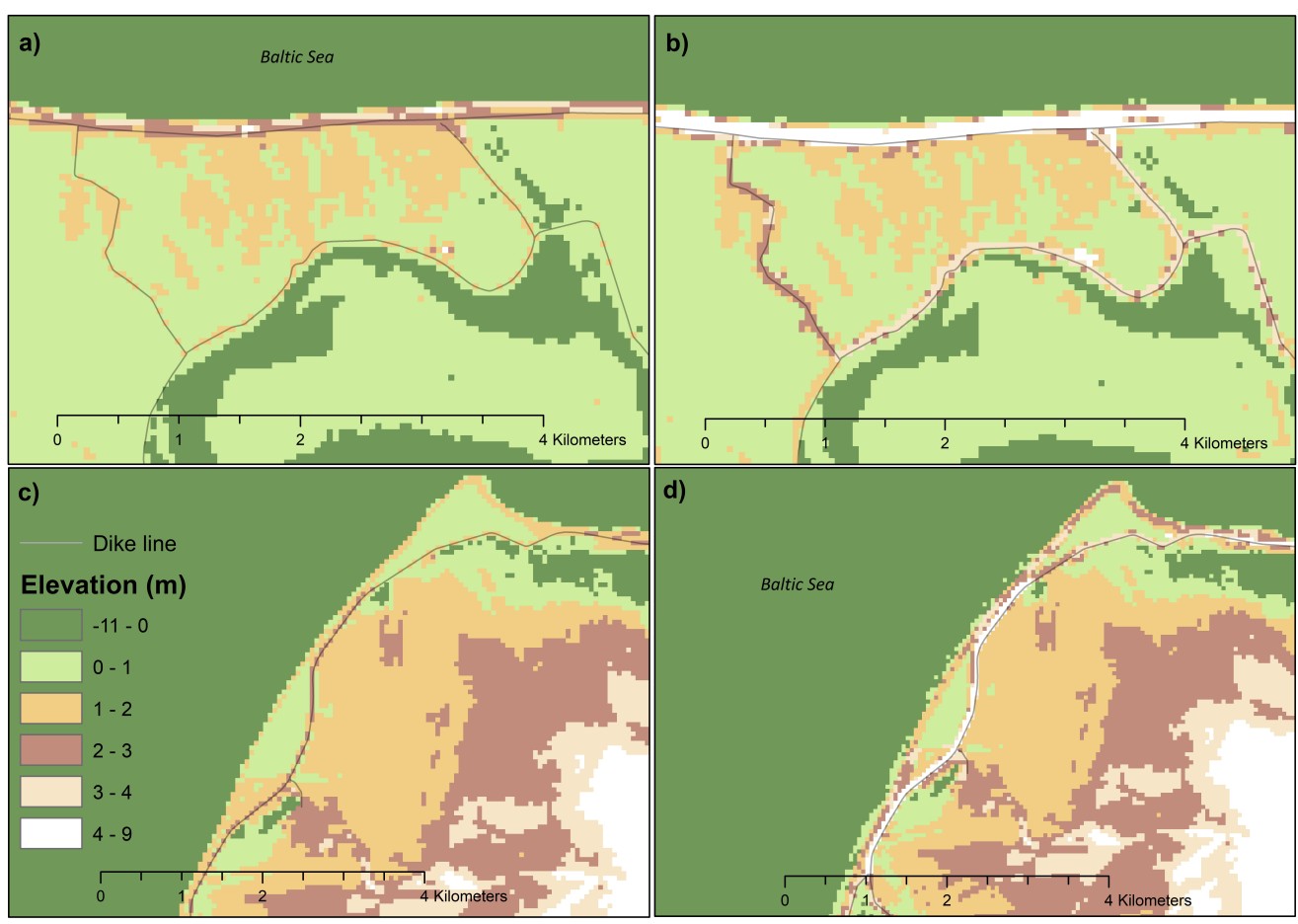

**Figure A1.** The difference between a DEM that was corrected for dikes (panels b) and d)) and an uncorrected DEM (panels a) and c)). A) and b) depict the area and dikes around Zingst (MP) and panels c) and d) show the northwestern coastline of the island of Fehmarn (SH).

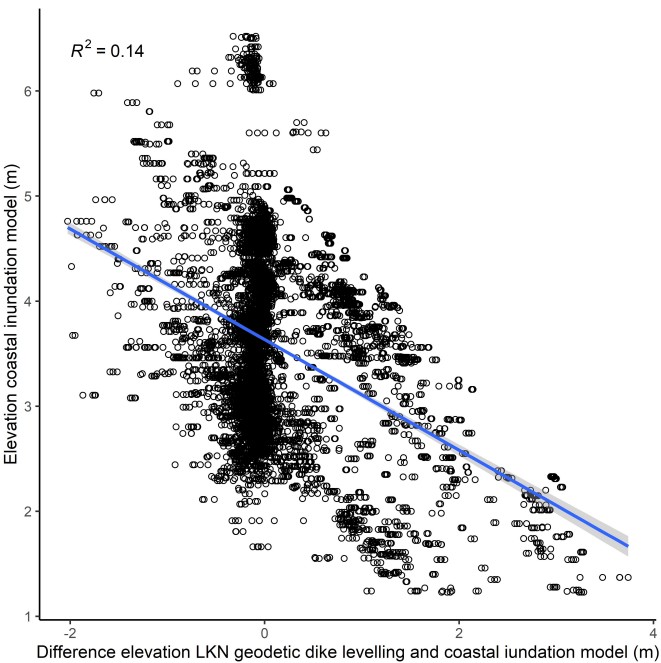

**Figure A2.** Correlation between the dike heights of the coastal inundation model and the difference between dike heights derived from the geodetic dike levelling (LKN) and dike heights of the coastal inundation model. The bue line represents the linear fit.

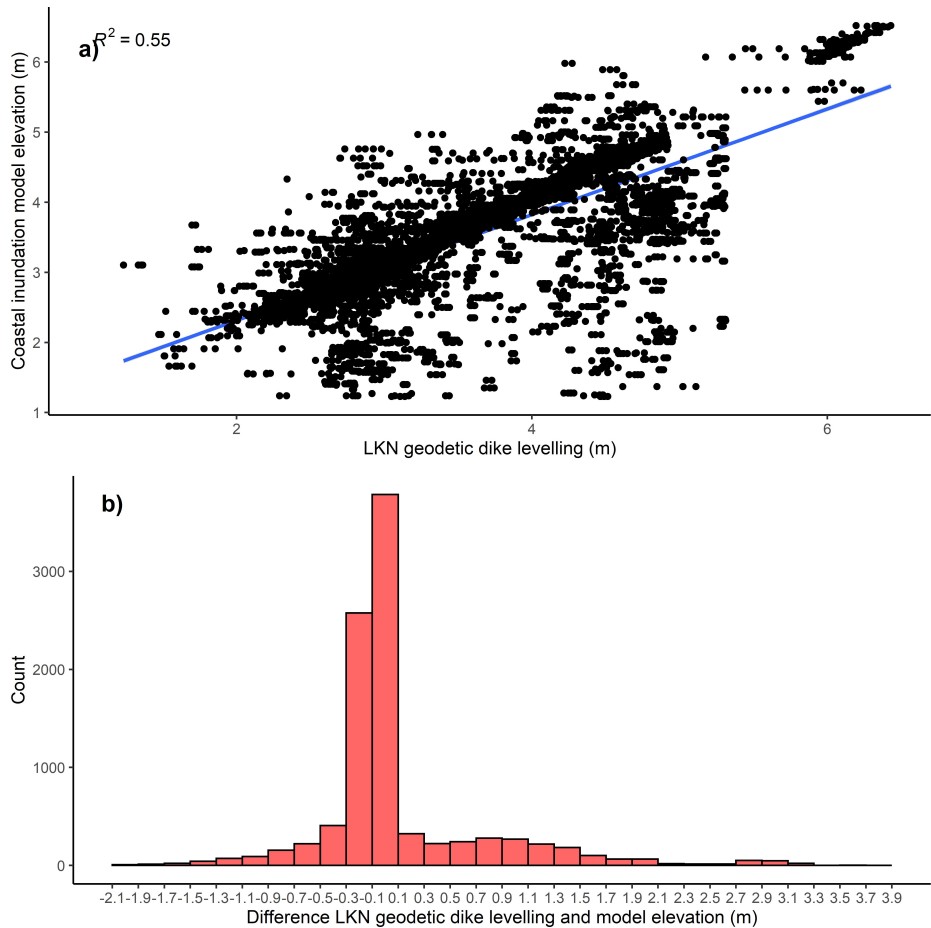

**Figure A3.** a) Correlation between the dike heights of the coastal inundation model and the dike heights derived from the geodetic dike levelling (LKN).b) Histogram showing the distribution of the error between the dike heights of the geodetic dike levelling and the coastal inundation model.

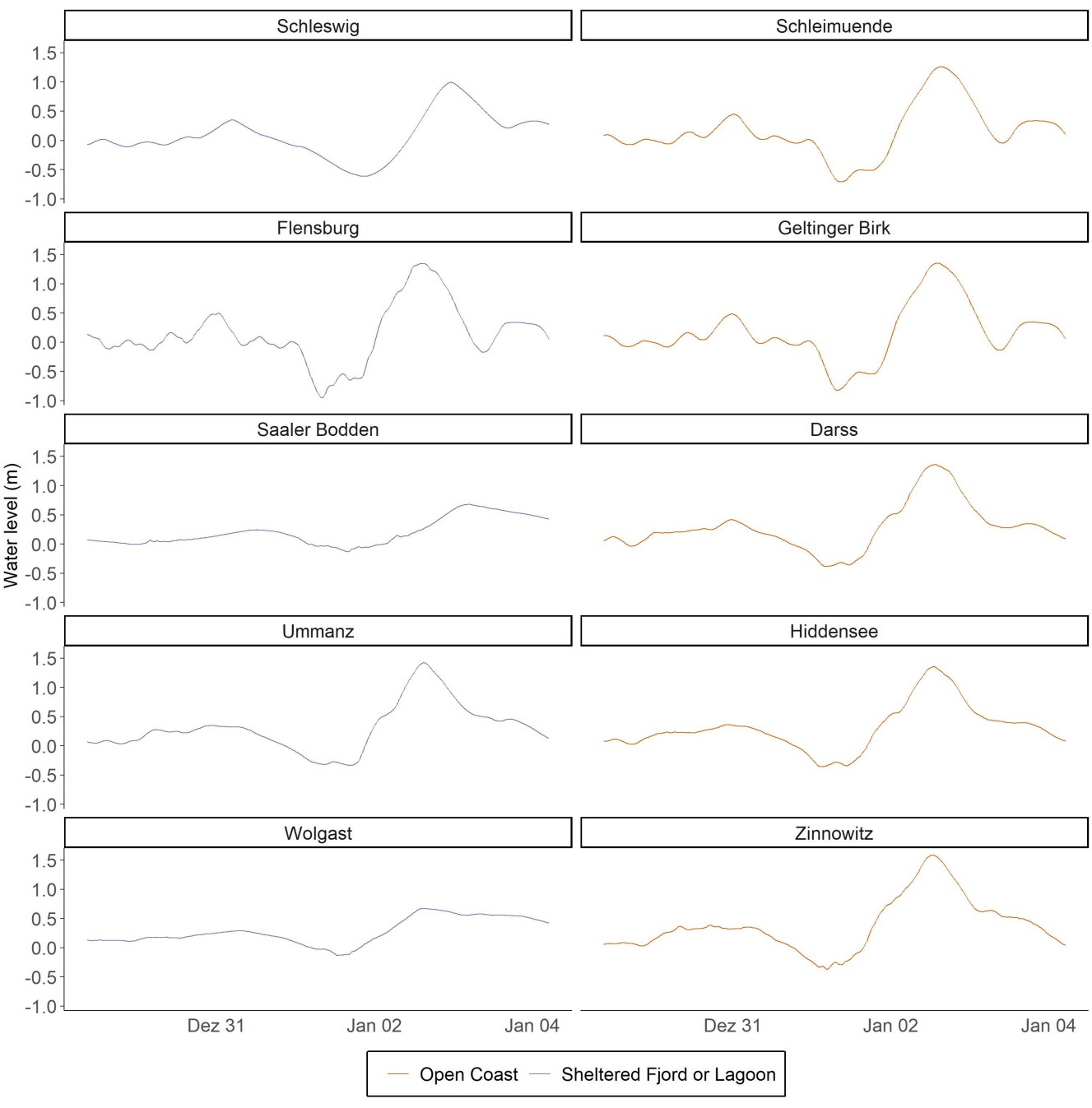

**Figure A4.** Water level timeseries for selected boundary stations during the January 2019 storm surge.

**Table A1.** Reclassification scheme of Corine land cover classes.

| Corine land class | New land class |
| --- | --- |
| continuous urban fabric | urban |
| discontinuous urban fabric | urban |
| industrial or commercial units | urban |
| road and railway networks and associated land | traffic |
| port areas | urban |
| airports | urban |
| mineral extraction sites | urban |
| dump sites | urban |
| construction sites | urban |
| green urban areas | green urban areas |
| sport and leasure facilities | green urban areas |
| non-irrigated arable land | agriculture |
| fruit tree and berry plantation | agriculture |
| pasture | agriculture |
| complex cultivation patterns | agriculture |
| land principally occupied by agriculture, with significant areas of natural vegetation | agriculture |
| broad-leaved forest | forest |
| coniferous forest | forest |
| mixed forest | forest |
| natural grassland | natural grassland |
| moors and heathland | wetland |
| transitional woodland-shrub | natural grassland |
| beaches, dunes and sand | unvegetated coastal sediment |
| sparsely vegetated areas | natural grassland |
| inland marshes | wetland |
| peat bog | wetland |
| saltmarshes | wetland |
| intertidal flats | unvegetated coastal sediment |
| water courses | inland waterbodies/courses |
| water bodies | inland waterbodies/courses |
| coastal lagoons | inland waterbodies/courses |
| estuaries | inland waterbodies/courses |
| sea and ocean | sea and ocean |

**Table A2.** Flood characteristics for varying configurations of Manning's *n* coefficients.

| | Domain 1 | | Domain 7 | | Domain 8 | |
|---|---|---|---|---|---|---|
| **Manning setup** | **flood extent (km$^2$)** | **mean flood depth (m)** | **flood extent (km$^2$)** | **mean flood depth** | **flood extent (km$^2$)** | **mean flood depth (m)** |
| Low | 5.78 | 0.47 | 24.36 | 0.39 | 33.21 | 0.58 |
| High | 5.69 | 0.46 | 22.04 | 0.37 | 32.17 | 0.57 |
| Land/Water | 5.79 | 0.47 | 24.12 | 0.38 | 33.4 | 0.58 |
| Uniform | 5.78 | 0.45 | 23.77 | 0.38 | 33.13 | 0.58 |
| Moderate | 5.73 | 0.46 | 22.84 | 0.38 | 32.58 | 0.58 |

*Author contributions.* JK, ATV, UG and ML designed the concept of the research. JK and ML set up the methods, ran the hydrodynamic models, analysed and plotted the results. MK processed and analysed the SAR imagery. JK wrote the manuscript with contributions from ML and MK. ATV, UG, ML and MK reviewed and edited the manuscript.

*Competing interests.* We declare that we have no competing interests.

*Acknowledgements.* JK would like to thank Dr. Jeffrey Neal (University of Bristol) and Dr. Sara Santamaria-Aguilar (University of Central Florida, previously Kiel University) for their support in setting up Lisflood. The authors would like to thank Prof. Dr. Horst Sterr (formerly Kiel University), Dr. Jacobus Hofstede (Scientific Director at Schleswig-Holstein State Government), Dr. Thomas Hirschhäuser (Head of the Hydrology Department at LKN) and Thorsten Dey (LKN) for providing helpful feedback on preliminary model outputs and fruitful discussions.

*Financial support.* This research is part of the ECAS-BALTIC project: Strategies of ecosystem-friendly coastal protection and ecosystem-supporting coastal adaptation for the German Baltic Sea Coast. The project is funded by the Federal Ministry of Education and Research (BMBF, funding code 03F0860H).

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
