# Peer review of "Regional assessment of extreme sea levels and associated coastal flooding along the German Baltic Sea coast"

_Natural Hazards and Earth System Sciences, 2022_

## Author Comment (AC1)

Revisions Manuscript:

*"A new modelling framework for regional assessment of extreme sea levels and associated coastal flooding along the German Baltic Sea coast"*

by Kiesel, J.; Lorenz, M.; König, M.; Gräwe, U.; Vafeidis, A.T.

https://doi.org/10.5194/nhess-2022-275

**Answers to reviewer #1**

We would like to thank anonymous referee #1 for the feedback and helpful comments concerning our manuscript. We have addressed all points (see our answers below) and added the aspect of glacial isostatic adjustment to section 2. (Study area).

Our detailed responses to every referee comment are listed below.

*The responses of the authors are written in green to enhance readability. All citations of text in the new, revised version of the manuscript are written in italics.*

Comments anonymous referee #1

1. Line 84 Explain what a hydrograph is, term might not be familiar to all readers

Revised as suggested. At the first occurrence of the term hydrograph, the text now reads:

*In order to account for spatial variations in water levels within each model domain, we defined a total of 32 'boundary stations' (Fig. 1), for which hydrographs (water level time series) were modelled in GETM for the four storm surge scenarios (2019 surge, 200-year event, 200-year event plus 1 m and 1.5 m).*

2. Line 167 Is the boundary condition the same for neighbouring boundary points? I understood from the text that the boundary point has as its boundary condition the hydrograph from the nearest boundary station. This could be explained in detail in the text.

Thank you for this remark. Reviewer #1 is right that neighbouring boundary points may have similar boundary conditions, as they receive their boundary conditions from the nearest flood boundary station (see Fig. 1). We have clarified this point in the text in section 3.2.3:

*Each boundary point received the boundary conditions (the hydrograph used to force the coastal inundation model in Lisflood) from the nearest boundary station (Fig. 1). In cases where the nearest flood boundary station to a boundary point located on the open coast was situated inside protected fjords or lagoons, or vice versa, we manually corrected that point to ensure that open coast boundary points are not forced with hydrographs of sheltered locations.*

3. Line 234 Has the land uplift due to postglacial rebound taken into account when SLR has been subtracted with a linear fit? You could mention the land uplift rate on the German Baltic coast and discuss whether it is relevant in this study.

Thank you for raising this important point. In contrast to the northern Baltic Sea, parts of the southern Baltic Sea, including the German Baltic Sea coast, are sinking as a consequence of postglacial rebound. Certainly, this is even more relevant for a modelling framework with the aim to assess coastal flood risk. However, subsidence in the southern Baltic Sea is generally variable and mostly well below 1 mm yr-1 but in places up to 2 mm yr-1 (Weisse et al. 2021; Richter et al. 2012). In addition, information is generally sparse across the study area and no consistent spatial information exists. For the above reasons and because subsidence is small compared to the extreme water levels and sea-level rise (IPCC 2021), we do not account for it in the present analysis. We have now clarified this in Section 2 (Study area) the manuscript:

Study area:

*In contrast to the emerging northern Baltic Sea coast, parts of the southern Baltic Sea coast are subsiding as a consequence of glacial isostatic adjustment. While subsidence is generally variable and mostly well below 1 mm yr-1, it can locally reach up to 2 mm yr-1 (Richter et al. 2012; Weisse et al. 2021). Due to the spatial variability, limited consistent information and rates mostly well below SLR, we have excluded subsidence from the present analysis.*

4. Table 4 and Figure 5:
It would be nice to have the station names of Table 4 in Figure 5 to be able to locate the stations in the map. The station could be given a number which is shown in Figure 5 to avoid too much text in the Figure. Figure 5 could also be larger, because it is one of the most interesting figures in the paper.

We added numbers to the TG stations in Tab. 3 and 4 and in former Fig. 5 and 6. We have added the identifying numbers and have changed the text accordingly. See also the response to comment 3.8 of reviewer 2.

5. Line 357 You could discuss why the peak water levels are higher in SH than in MP in the 200-year event. Is it due to the shape of the coastline, does the bathymetry affect it?

Thank you for the comment. We have added some text explaining why the surges are higher in SH than MP. The simple reason is that the fetch length, i.e. the distance over which wind stress can affect the water level undisturbed, is larger for SH than MP (e.g. the west-east axis of water). The added text reads:

*The spatial distribution of the 30-year and 200-year return levels (Fig. 5) shows that the highest ESLs occur at the coasts, with decreasing ESLs from west to east. The pattern and return levels have*

*already been described in the literature (Gräwe and Burchard 2012; Wolski et al. 2014), and are primarily due to fetch length, which is longer for the coast of Schleswig-Holstein (SH).*

We have furthermore rewritten the entire paragraph and have improved the discussion with respect to the differences in mean inundation depth and peak water levels. This section now reads:

*Our results show that the differences in flood depth between both states are due to variations in peak water level and coastal morphology. For the 200-year event, peak water levels are on average 0.46 m lower in MP compared to SH (2.03 m in SH and 1.57 m in MP, Table 4). In both SLR-scenarios, flood depth is slightly higher in MP, which we attribute to the substantially larger flood extent and lower elevations within the floodplain. Within the flood extent of the 200-year event and 1.5 m of SLR, mean elevation is 1.1 m NHN in MP compared to 1.27 m NHN in SH.*

**Publication bibliography**

Gräwe, Ulf; Burchard, Hans (2012): Storm surges in the Western Baltic Sea: the present and a possible future. In *Clim Dyn* 39 (1-2), pp. 165–183. DOI: 10.1007/s00382-011-1185-z.

IPCC (2021): Climate Change 2021: The Physical Science Basis. Contribution of Working Group I to the Sixth Assessment Report of the Intergovernmental Panel on Climate Change. Edited by V. Masson-Delmotte, P. Zhai, A. Pirani, S. L. Connors, C. Péan, S. Berger et al. Intergovernmental Panel on Climate Change. Cambridge University Press, Cambridge, United Kingdom and New York, USA. Available online at doi:10.1017/9781009157896.

Richter, A.; Groh, A.; Dietrich, R. (2012): Geodetic observation of sea-level change and crustal deformation in the Baltic Sea region. In *Physics and Chemistry of the Earth, Parts A/B/C* 53-54, pp. 43–53. DOI: 10.1016/j.pce.2011.04.011.

Weisse, Ralf; Dailidienė, Inga; Hünicke, Birgit; Kahma, Kimmo; Madsen, Kristine; Omstedt, Anders et al. (2021): Sea level dynamics and coastal erosion in the Baltic Sea region. In *Earth Syst. Dynam.* 12 (3), pp. 871–898. DOI: 10.5194/esd-12-871-2021.

Wolski, Tomasz; Wiśniewski, Bernard; Giza, Andrzej; Kowalewska-Kalkowska, Halina; Boman, Hanna; Grabbi-Kaiv, Silve et al. (2014): Extreme sea levels at selected stations on the Baltic Sea coast. In *Oceanologia* 56 (2), pp. 259–290. DOI: 10.5697/oc.56-2.259.

---

## Author Comment (AC2)

Revisions Manuscript:

*"A new modelling framework for regional assessment of extreme sea levels and associated coastal flooding along the German Baltic Sea coast"*

by Kiesel, J.; Lorenz, M.; König, M.; Gräwe, U.; Vafeidis, A.T.

https://doi.org/10.5194/nhess-2022-275

**Answers to reviewer #2**

We would like to thank anonymous referee #2 for the effort and the very detailed evaluation of our manuscript. We found the comments constructive and helpful for improving the manuscript.

In response to R2´s comments, we have rearranged the introduction and now state more clearly the objectives of our paper, which are now: *"1) exploring how flood extent may change until the end of the century, if existing dikes are not upgraded, by applying two high-end SLR scenarios (1 m and 1.5 m); 2) identifying hotspots of coastal flooding in the study region and 3); evaluating the use of SAR-imagery for validating the simulated flood extents."* In addition, we have made clarifications in the methods section with respect to the extreme value analysis and the coastal ocean model. We have also conducted a new analysis, where we compared the elevation variability within the flood extent of the 200-year event and 1.5 m of SLR between both federal states. Last, we have extended our discussion on the potential effects of neglecting surge hydrograph variability (i.e. surge intensities/durations) on flood characteristics and the uncertainties around the extrapolation of the 200-year return water levels. In addition, we suggest a new title for our manuscript: *"Regional assessment of extreme sea levels and associated coastal flooding along the German Baltic Sea coast".*

Our detailed responses to every referee comment are listed below.

*The responses of the authors are written in green to enhance readability. All citations of text in the new, revised version of the manuscript are written in italics.*

**Comments anonymous referee #2**

The manuscript describes a modelling framework for the German Baltic coast to assess coastal extreme sea-levels and associated flooding. Such a tool is of great utility for coastal management and planning in view of sea-level rise. The methods employed are within the state of the art (numerical modelling tools, analysis methods for EVA, etc) . The framework is not new in terms of these methods (multiple similar systems have been developed in other parts of the world) but it is new and the first of it's kind in the specific region of the German Baltic coast, according to the authors. As such, it provides the opportunity to bring new knowledge on coastal flood risk in this region. However, I think the manuscript should be re-structured to better highlight its objetive. It appears that the manuscripts aims to describe and

validate the modelling framework but also demostrate its use for a specific application, which seems to be coastal protection design (given the recurrent link to the 200 year RP design level and the simulations performed for this RP). Or it could be that the objective is mainly the design level calculations and associated flood charachteristics, and the validation is just a necessary step? This is just my interpretation, but whether it is correct or not, the message is that the objective is not clearly presented in the introduction and followed through in the rest of the document. For example, a proper carachterization of flood charachteristics - which is what the title of the manuscript implies -would require the assessment of multiple return periods, while here results focus (only) on the 200 RP design level. I therefore recommend to restructure the manuscript, and rename it if necessary, according to the objetive and research question at hand. This objetive should be clearly stated in the introduction, before elaborating on the methods used and simulations performed (which should be justified by the chosen objective), and it should streamline the rest of the document. Because of the need to restructure the contents, I propose a major revision. That said, the manuscript is scientifically sound, so no new results should be needed to get it to a publishable level.

We thank R2 for the detailed feedback and thoughts on our manuscript. We have reframed our research question in the introduction and adjusted the discussion and conclusion sections accordingly. Please find the details of our adjustments in our answers to your specific comments below.

1.

Additionally, harmonization is needed relative to the use of the terms waterlevel, surge etc. This relates to some specific comments pointed out below. As it is, it is not clear in the document if the return periods derived correspond to waterlevels (including tides) or just surges. The term waterlevel is mostly employed, but according to the text the hydrograph definition is based on surge. This aspect should be clarified in the methods.

Thank you for the comment. Since the Baltic Sea is characterised by a microtidal regime, the effects of tide-surge interactions on total water levels are negligible (Arns et al. 2020). In order to clarify this, we have added information on the tidal amplitude in section 2, which now reads:

*The Baltic Sea is characterised by a microtidal regime (tidal range varying between 0.1 m and 0.2 m (Sterr, 2008), low salinity, strong stratification, and anoxic conditions in many areas (Meier et al., 2022).*

In addition, we clarify in section 3.4 that return water levels correspond to surges only (equal to total water level), as tides are negligible in the study region. Please see our suggestion for revision:

*Due to the microtidal regime of the Baltic Sea, the derived return periods and water levels correspond only to the surge component, as tidal contributions to ESL (tide-surge interactions) is negligible* (Arns et al. 2020).

2. **Introduction**

**2.1**

A list of difficulties faced during "large" scale coastal flooding assessments is presented (resolution and computational burden, complexity and sensitivity of models, lack of accurate

data on flood protection features like dunes and dikes, validation material, uncertainty in extrapolation to high RPs, etc), and it is followed by 'Here we address these challenges by developing a new modelling framework' (line 74). Again, this is not the objetive of the paper, as it doesn't provide a solution to lack of computational resouces to reach relevant flood scales at regional scales (for example showing a faster modelling framework) , sparseness of data, etc. I think this part of the introduction should be rephrased to highlight that these typical difficulties, which hamper the validity of broad scale flood assessments, have been taken into account in the design of the current modelling framework, by collecting data on hard structures, using a high-resolution DEM, etc, which were fortunately accessible to the authors for this specific region. These ascpects remain a challenge elsewhere. In terms of validation, the usability of SAR is explored as a promising (and emerging) data source.

We thank R2 for this helpful comment. We agree that this paper does not address the named limitations in a way that would help the development of such modelling frameworks in other, data scarce regions. We agree that we only address these problems within our modelling framework for our specific study region. Therefore, we have adjusted the wording in the text and now state more clearly the objectives of our paper before we elaborate about the methods used. Please find our specific edits below:

First, we have added a sentence at the beginning of the paragraph in the introduction, describing the difficulties and limitations of coastal inundation modelling:

*State-of-the-art coastal flood maps should consider oceanographic forcings, projected SLR, detailed topographic data on coastal morphology, including anthropogenic coastal protection measures such as dikes, and the effects of land cover on flood propagation.*

Second, we point out in the introduction that studies that have used state-of-the-art hydrodynamic modelling (considering temporal evolution of the surge and the effects of surface roughness) to assess coastal flooding along the entire German Baltic Sea coast are still missing.

*Along the German Baltic Sea coast, existing studies on coastal flooding have either used state-of-the-art hydrodynamic models, but cover only a small fraction of the study region* (Höffken et al. 2020; Vollstedt et al. 2021)*, or assess potential flood extents for the entire region, but rely on global topographic data sources and apply the bathtub approach* (Schuldt et al. 2020)*. In addition, the validation of produced flood extents is not provided.*

*There is a need to simulate coastal flooding on a regional scale, considering the limitations of large-scale coastal flood mapping mentioned before. This is particularly true for topographic data sources and the incorporation of coastal protection infrastructure, which constitute the main bottlenecks for the quality of coastal flood risk assessment (Vousdoukas et al. 2018).*

Finally, we have rephrased the objective of the paper, now emphasizing that we aim to assess coastal flooding in the study region for an event that is equivalent to existing design heights for state embankments:

*Here, we simulate coastal flooding along the German Baltic Sea coast for a storm surge that aligns with the design standard of state embankments in the region, i.e.\ the 200-year return water level. This study aims at: 1) exploring how flood extent may change until the end of the century, if existing dikes are not upgraded, by applying two high-end SLR scenarios (1 m and 1.5 m); 2) identifying*

*hotspots of coastal flooding in the study region, and 3); evaluating the use of SAR-imagery for validating the simulated flood extents. To the knowledge of the authors, this study constitutes the first regional-scale assessment using a high resolution, fully validated, and offline-coupled hydrodynamic modelling framework that incorporates natural and anthropogenic flood barriers to assess extreme sea levels and associated coastal flooding along the German Baltic Sea coast.*

**2.2**

If the manuscript is reframed to target the dike/coastal protection design application (even if as a showcase for the framework), I propose that the associated, area-specific information presented in section 2 (lines 107-113) is moved to the introduction as part of the contextualization of the question and objetive at hand.

Thank you for this helpful suggestion. We have indeed moved this section into the introduction. Please see our suggestion for revision in the new version of the MS, which now reads:

*The German Baltic Sea coast comprises a total length of 2538 km and 27 % of this coast is protected by dikes* (Sterr 2008; van der Pol et al. 2021). *Today, state embankments in both federal states, Schleswig-Holstein (SH) and Mecklenburg Western Pomerania (MP) are designed to be high enough to prevent flooding during a storm surge with a return period of 200 years plus a buffer of 0.5 m to account for SLR* (StALU 2012; Melund 2022). *On the other hand, regional dikes have a variable but generally lower standard of protection* (Melund 2022). *During the last two decades, the concept of so-called climate dikes has been introduced as a paradigm in embankment construction. The climate dike accounts for uncertainties related to SLR projections by having wider dike crests, which allows for a comparatively easy and rapid increase in dike heights without reconstructing the dike base. The climate dike can easily be increased in height by up to 1.5 m, and suggested extension options include 0.5 m and 1.0 m* (Melund 2022).

**3. Methods**

**3.1**

Line 124: Please include a proper reference to the IPCC report mentioned.

Reference to IPCC is now included.

**3.2**

Line 132: correct hind-cast to hindcast

Revised as suggested.

**3.3**

General comment in section 3.2.1: Information on the modelled physical processes is missing, does the model include tides (at the boundary)? What atmospheric parameters are used for forcing (wind is mentioned, but what about atmospheric pressure?) Also, 2 different forcing products are used for hindcast and for the 2019 event, what are the spatial/temporal resolutions of these products? (as this has a mayor impact on storm-surge extremes). Some calibration of bottom friction is mentioned in order to increase extreme surge heights, and hindcast winds have been increased also to increase the surge. Little information on these calibration exercises is provided, could you elaborate a bit more? It is important to refer to this in the discussion,as it is mentioned that waves setup was not included because the model showed already an overprediction of extremes.…

We have only used 10m wind speeds and sea level pressure. The temporal resolution of the data is hourly at ~11km resolution (UERRA) and 3-hourly at ~7km resolution (DWD), respectively. We added this information to the main text. The 7% increase is needed to improve the negative bias of extreme sea levels of the model compared to the tide gauge data. The 7% are a result of a study conducted by Lorenz and Gräwe (2023), where an ESL hindcast ensemble of the whole Baltic Sea region is produced. Still, the model generally underestimates ESLs for most tide gauges, except one event that the model overestimates, see the GEV for KielHoltenau, former Fig. 4. The discussion on wave setup has been rewritten (see our response to your comments 4.12 and 4.13).

**3.4**

Section 3.3 -->Rename to 'Sensitivity analysis'. A calibration would entail benchmarking against the truth (e.g. observations) to choose the best settings, but this is not done.

We agree and have adjusted the section title accordingly.

**3.5**

Section 3.4 --> There is too much detail on the GEV method (e.g. thedescription of subfamilies), a proper literature reference should suffice (as it is a very widely used method). The generation of the hydrographs constitutes a more novel component here, consider adding in this section an example for the generation of the hydrograph and the scaling to a given RP. In relation to the generation of the hydrographs (Appendix A), please include a justification for the choice of 1meter as threshold to isolate extreme events. In typical peak-over-threshold (PoT) extreme value sampling, a high percentile is often used, which ensures a minimum sample size. Do you use meteorological independence criteria between events (as typically done in PoT)? Is the 6 day event duration (3 days before, 3 days after) justified somehow for this region?

We shortened the GEV section by removing reference to the three special cases and the Figure showing these (former Fig. 2). On the other hand, we moved the section on the generation of the mean surge shape from the Appendix to the Method Section. The choice of 1m is the official classification of a storm surge along the German Baltic Sea coast by the German Federal Maritime and Hydrographic Agency. We adopted this definition to make interpretation of this study as easy as possible to local authorities. This information has also been added to the text. The choice of the interval of plus/minus 3 days is chosen as it covers the whole ESL event plus some spin-up time for the inundation model, which can be seen from the constructed mean surge shapes, former Fig. A1.

**3.6**

Line 225: replace 'stretched' with 'scaled'

Revised as suggested.

**3.7**

Line 226: replace 'than' with 'then'

Revised as suggested.

**3.8**

Section 3.5 --> It would be good to include a spatial map of the TGs used, with a number asigned to each, such that these identifiers can be used throughout the body of the text (for example when mentioning a TG, with name[number] ) and in subsequent plots. It is difficult to follow otherwise for someone not familiar with the region. For the EVA, it would be interesting to see for those TGs where your records are longer than in your hindcast, what is the impact of clipping the TG series to the hindcast period on the EVA? You could be sampling more (and larger/smaller) extreme events in your TG than in your model...which might (partly) explain differences in the fits.

We added numbers to the TG stations in Tab. 3  and 4 and in former Fig. 5 and 6. We have added the identifying numbers and have changed the text accordingly. Regarding the EVA, we find no significant changes of the statistics for most stations, when clipping the TG time series to the model period, see Table 1, except for the very long time series. For TG 'Gedser' (starting in 1891), the return levels are significantly reduced by ~12 cm (30-year RP) and ~20cm (200-year RP), when only considering the overlap with the model. For 'Hornbaek' that also starts in 1891, the difference is ~2 cm (30-year RP) and ~8 cm (200-year RP), when only considering the overlap. We added the results of these two TGS as two additional rows to Tab. 4. The model fits better to the GEV based on the full time series of the TG.

Table 1: List of the return levels (RL) of the 30-year and 200-year return periods including the 95 % confidence intervals for different tide gauges along the Baltic Sea coast estimated from observations (obs.) and model results (mod.). For the observations we included here columns which only use the data which overlaps with the time period of the model.

| Number | station | 30-year obs full | 30-year obs overlap | 30-year model | 200-year obs full | 200-year obs overlap | 200-year model |
|---|---|---|---|---|---|---|---|
| 0 | Althagen | 0.92 ± 0.24 | 0.89 ± 0.09 | 1.36 ± 0.19 | 1.10 ± 0.36 | 1.02 ± 0.17 | 1.71 ± 0.44 |
| 1 | Barhoeft | 1.35 ± 0.08 | 1.32 ± 0.13 | 1.33 ± 0.11 | 1.44 ± 0.12 | 1.45 ± 0.18 | 1.53 ± 0.13 |
| 2 | Barsebaeck | 1.31 ± 0.13 | 1.10 ± 0.02 | 0.91 ± 0.08 | 1.45 ± 0.19 | 1.11 ± 0.02 | 1.00 ± 0.16 |
| 3 | Eckernfoerde | 1.69 ± 0.21 | 1.69 ± 0.22 | 1.74 ± 0.24 | 1.91 ± 0.39 | 1.93 ± 0.41 | 2.05 ± 0.39 |
| 4 | Flensburg | 1.63 ± 0.11 | 1.63 ± 0.13 | 1.72 ± 0.16 | 1.80 ± 0.21 | 1.82 ± 0.25 | 2.03 ± 0.29 |
| 5 | Gedser | 1.51 ± 0.08 | 1.39 ± 0.09 | 1.51 ± 0.13 | 1.71 ± 0.11 | 1.51 ± 0.15 | 1.74 ± 0.26 |
| 6 | Greifswald | 1.49 ± 0.12 | 1.49 ± 0.12 | 1.49 ± 0.12 | 1.65 ± 0.22 | 1.66 ± 0.23 | 1.68 ± 0.23 |
| 7 | Heiligenhafen | 1.72 ± 0.35 | 1.73 ± 0.36 | 1.61 ± 0.17 | 2.15 ± 0.83 | 2.16 ± 0.86 | 1.90 ± 0.35 |
| 8 | Hornbaek | 1.49 ± 0.06 | 1.47 ± 0.06 | 1.43 ± 0.13 | 1.64 ± 0.07 | 1.56 ± 0.07 | 1.60 ± 0.25 |
| 9 | Kappeln | 1.36 ± 0.13 | 1.37 ± 0.14 | 1.66 ± 0.18 | 1.51 ± 0.23 | 1.52 ± 0.25 | 2.01 ± 0.34 |
| 10 | KielHoltenau | 1.73 ± 0.18 | 1.74 ± 0.19 | 1.73 ± 0.16 | 1.99 ± 0.36 | 2.00 ± 0.38 | 2.03 ± 0.29 |
| 11 | Klagshamn | 1.13 ± 0.08 | 1.16 ± 0.09 | 1.20 ± 0.10 | 1.27 ± 0.16 | 1.28 ± 0.18 | 1.37 ± 0.21 |
| 12 | Koserow | 1.42 ± 0.05 | 1.42 ± 0.05 | 1.38 ± 0.14 | 1.47 ± 0.07 | 1.47 ± 0.07 | 1.56 ± 0.29 |
| 13 | Langballigau | 1.66 ± 0.26 | 1.67 ± 0.28 | 1.69 ± 0.16 | 1.93 ± 0.56 | 1.96 ± 0.62 | 1.99 ± 0.28 |
| 14 | LTKalkgrund | 1.67 ± 0.29 | 1.69 ± 0.36 | 1.67 ± 0.15 | 2.03 ± 0.65 | 2.07 ± 0.89 | 1.96 ± 0.27 |
| 15 | Neustadt | 1.64 ± 0.22 | 1.65 ± 0.23 | 1.69 ± 0.17 | 1.85 ± 0.43 | 1.87 ± 0.46 | 1.99 ± 0.33 |
| 16 | Rostock | 1.55 ± 0.19 | 1.55 ± 0.19 | 1.52 ± 0.15 | 1.83 ± 0.39 | 1.84 ± 0.40 | 1.75 ± 0.31 |
| 17 | Sassnitz | 1.22 ± 0.08 | 1.23 ± 0.08 | 1.24 ± 0.11 | 1.33 ± 0.14 | 1.33 ± 0.14 | 1.44 ± 0.21 |
| 18 | SchleimundeSP | 1.62 ± 0.24 | 1.63 ± 0.26 | 1.69 ± 0.15 | 1.88 ± 0.49 | 1.91 ± 0.55 | 1.98 ± 0.26 |
| 19 | Schleswig | 1.39 ± 0.11 | 1.39 ± 0.10 | 1.91 ± 0.28 | 1.48 ± 0.21 | 1.47 ± 0.18 | 2.44 ± 0.65 |
| 20 | Simrishamn | 1.09 ± 0.09 | 1.09 ± 0.09 | 1.19 ± 0.11 | 1.19 ± 0.15 | 1.19 ± 0.15 | 1.37 ± 0.23 |
| 21 | Skanor | 1.27 ± 0.21 | 1.27 ± 0.18 | 1.31 ± 0.10 | 1.46 ± 0.44 | 1.44 ± 0.37 | 1.48 ± 0.19 |
| 22 | Stralsund | 1.41 ± 0.13 | 1.42 ± 0.13 | 1.39 ± 0.11 | 1.59 ± 0.23 | 1.59 ± 0.24 | 1.58 ± 0.21 |
| 23 | Timmendorf | 1.65 ± 0.18 | 1.66 ± 0.18 | 1.65 ± 0.18 | 1.93 ± 0.38 | 1.92 ± 0.36 | 1.94 ± 0.38 |
| 24 | Travemunde | 1.71 ± 0.13 | 1.66 ± 0.15 | 1.72 ± 0.18 | 1.92 ± 0.24 | 1.86 ± 0.28 | 2.03 ± 0.36 |
| 25 | Ueckermuende | 1.00 ± 0.12 | 1.00 ± 0.12 | 0.90 ± 0.13 | 1.15 ± 0.25 | 1.15 ± 0.24 | 1.11 ± 0.30 |
| 26 | Viken | 1.49 ± 0.16 | 1.49 ± 0.17 | 1.32 ± 0.13 | 1.69 ± 0.31 | 1.71 ± 0.33 | 1.49 ± 0.25 |
| 27 | Warnemuende | 1.54 ± 0.16 | 1.51 ± 0.17 | 1.50 ± 0.15 | 1.81 ± 0.35 | 1.76 ± 0.37 | 1.74 ± 0.32 |
| 28 | Wismar | 1.69 ± 0.13 | 1.70 ± 0.15 | 1.68 ± 0.19 | 1.89 ± 0.23 | 1.91 ± 0.26 | 1.97 ± 0.40 |
| 29 | Wolgast | 0.96 ± 0.11 | 0.96 ± 0.09 | 0.95 ± 0.13 | 1.09 ± 0.23 | 1.08 ± 0.12 | 1.21 ± 0.33 |

**4. Results and discussion**

**4.1**

Jusitfy the choice of 1meter as threshold for extremes in the hindcast validation (as for the event isolation in Appendix A, see comment before)

See comment 3.5.

**4.2**

Near coastal lagoons, the model has been deemed inapropriate and TGs have been used instead for the flood modelling. This might be justified for this particular application or assessment, as an ad-hoc solution, but it is not part of the framework per se. This should be discussed in section 4.6.

Thank you for this point. We have now added a paragraph to section 4.6 that discusses this limitation of our modeling framework:

*We find that the coastal ocean model, which provides the boundary conditions for the coastal inundation model, overestimates both the extrapolated ESL (the modelled 200-year return water level) and the water levels of the 2019 surge inside protected lagoons of Fischland-Darß-Zingst in MP and The Schlei in SH (Fig. 1, Fig. 6, Tab. 4). We have therefore used ESL extrapolations from the tide gauges Althagen (0), Kappeln (9) and Schleswig (19) as input for the coastal inundation model. We note that this change is only valid for this specific application and should not be considered part of the modelling framework. Thus, the provision of boundary conditions from coarser resolution hydrodynamic models (e.g. Global Tide and Surge Model* (Muis et al. 2020)*) for sheltered coastal environments constitutes an extra source of uncertainty in large-scale coastal flood risk assessments that should be accounted for in future applications.*

**4.3**

In line 210, 30 tide-gauges are listed, but only 28 appear in Figure 6. Why?

These gauges lie outside (north) of the shown domain.

**4.4**

The empirical values in the distribution shown for a TG (Figure 4) show an increasing bias for increasing RPs, but the fit seems insensitive to this. Could you elaborate? This seems also evident from the 2019 event, what is the corresponding RP and how does the error compare to the derived RP error, given that a different meteo forcing is used? The mean (negative) bias for the 2019 event is partly compensated by the positive bias in the stations around some

of the unresolved lagoons, you could leave these out to have a better picture of the model performance for this event? (since TGs are used for the flooding at these locations anyways)

Yes, the model has generally a negative bias, especially the for the high ESLs. However, by overestimating the highest event in the model time period, the statistical fits of the model and the observations are very close, see also the return levels in Tab. 4. For some tide gauge locations, the 200-year return levels from the model are higher than return levels obtained from the observation based GEV. This is especially true for the coast of Schleswig-Holstein, where one event is overestimated by the model, thus influencing the tail of the distribution. However, the 95% confidence intervals are large due to the extrapolation and both, the estimated mean return levels of the model and the observation fall into the confidence intervals of each other.

We have compared the observed/measured (TG) return periods for the 2019 event (blue numbers in Fig. 1) with the return periods calculated with our model (black in Fig. 1). We find that the modelled values are within the respective confidence interval for most locations.

However, the estimation of return periods based on the discrete observations (explanation of the term "discrete observation" provided in caption of Fig. 1) are much smaller for many locations compared to the respective RP from the GEV fits (red numbers for GEV based on observations, black numbers for GEV based on model results). We cannot compare the errors here to other meteorological forcings since we have only run this model for the UERRA dataset. However, for the entire Baltic Sea, Lorenz and Gräwe (2023) show that for a hindcast ensemble of 13 members at ~1.8 km resolution that the ensemble spread of annual mean maximum water levels is in the order of the variability between the meteorological datasets for the 99[th] percentile wind speeds.

We prefer to not dismiss the lagoons from the comparison, but rather show the limitations of the model resolution.

[Figure]

Figure 1: Return levels and 95% confidence intervals based on the observation GEV (red), model GEV (black), and for the discrete return periods of the 2019 event. Discrete means that in case the 2019 event is the highest for a 50-year long data record, the RP is 50 years. If it would be the second highest, the discrete RP would be 25, and so on. The respective return periods from the GEV with a return level as high as the 2019 event are listed in the respective colors above. These RP are usually larger than the discrete ones.

**4.5**

For the Hit/False/Miss score assessment, could you not interpolate the 50m modelled fields to the SAR resolution of 10 meters? The score percentages are confusing as they are

Thanks for this remark. We agree that these numbers are confusing and have therefore now converted both the SAR and LISFLOOD-FP flood extents to polygons. Using the polygon for calculating the indices, we now came up with 50 % of the inundated area as being correctly predicted and 50 % missed (inundation observed in the SAR-imagery but not in the coastal inundation model). In addition, 50 % of the inundated area as simulated with the coastal inundation model was overpredicted. We have also improved and extended the discussion around the limitations of our flood extent validation, which we now frame as comparison.

**4.6**

Lines 355-359: There is a lot of speculation in these lines. Multiple instances of 'We believe that...'. Avoid speculating, you have all the results at hand to do an analysis of what is leading to differences between regions.

We agree and have rephrased this paragraph. In addition, we conducted a new analysis where we compared the elevation variability within the flood extent of the 200-year event and 1.5 m of SLR between both federal states. This passage now reads:

LXY

*Our results show that the differences in flood depth between both states are due to variations in peak water level and coastal morphology. For the 200-year event, peak water levels are on average 0.46 m lower in MP compared to SH (2.03 m in SH and 1.57 m in MP, Table 4). In both SLR-scenarios, the flood depth is slightly higher in MP, which we attribute to the substantially larger flood extent and lower elevations within the floodplain. Within the flood extent of the 200-year event and 1.5 m of SLR, the mean elevation is 1.1 m NHN in MP compared to 1.27 m NHN in SH.*

**4.7**

Line 370: I find this discussion bizarre, and the purpose of analyzing the linear correlation with peak WL is not explained. Why would we have a linear increase in flooded area? That depends on the inland topography, the hydraulic connectivity etc, no? why are you showing this?

We agree with your comment and have therefore deleted this paragraph and the figure from the manuscript.

**4.8**

Line 382:*To the knowledge of the authors, this study constitutes the first regional-scale assessment using a high resolution, fully validated and offline-coupled modelling framework that incorporates natural and anthropogenic flood barriers to assess extreme sea levels and associated coastal flooding along the German Baltic Sea coast.* This should be in the introduction, together with a clear objective.

We have moved this sentence to the introduction and clarified the objectives of the paper according to comment 2.1.

**4.9**

Line 388: *Gulf of Finland and Florida-->*It is a strange mix of locations, I suggest to remove Florida and focus on Finland which is at least in the Baltic.

Done. This section now reads:

*For example, in a study from the Gulf of Finland, the contribution of wave setup to extreme water levels approached up to 50 %, while the maximum absolute contribution to peak water levels varied between 70 cm and 80 cm in exposed areas* (Soomere et al. 2013).

**4.10**

Line 390: 'in the previous location' rephrase it is not clear which you refer to.

Done. Please see answer to previous comment.

**4.11**

Line 394: : *first, wave setup is included in 395 the tide gauge records that we use to extrapolate the 200-year return water levels.* This is a bold statement, up until recently it was argued that TGs didn't record wave setup because of their sheltered location in harbors, but recent studies have shown that sometimes this assumption doesn't hold. Do you have prove or can you reference a paper where this is evaluated for this region? Otherwise rephrase.

We completely rewrote the discussion on wave setup, see our response to your comment 4.12 and 4.13.

**4.12**

Line 395: *for 21 of the 30 tide gauges, the coastal ocean model still overestimates the 200-year return water level.* Elaborate on why this is the case, while for the 2019 we have a general underestimating trend, and it has been shown that extremes are generally underestimated in Fig 3. It looks like the EVA plays a role here? The model was calibrated also to increase surges (increased wind, reduced bottom friction), still it seems to generally underestimate, but after the GEV fit it overestimates? Using this same argument, and if one only looked at the hindcasted extremes (actual modelled extremes), one could advocate to add wave-setup instead in order to increase the underestimated surges...

Yes, some 200-year return levels from the model are higher than return levels obtained from the observation based GEV. This is especially true for the Schleswig-Holstein coast where one event is significantly overestimated by the model (see former Fig. 4). This one event influences the tail of the distribution, which is especially important for the extrapolation to 200 years. However, the 95% confidence intervals are large due to the extrapolation and both, the estimated mean return levels of the model and observation fall into the confidence intervals of each other. See also Point 4.4.

Yes, wave setup may play a role here, however, other uncertainties could partly explain the bias: the meteorological dataset may not represent the storms that cause the highest water levels correctly. E.g. the resolution may not the high enough to properly resolve the land-sea mask in the reanalysis model or the model simply has biases.

We added these points now in the discussion in the new version of the manuscript. This part of the discussion now reads:

*We still excluded wave setup in our analysis. A technical reason is that the coastal ocean model has a resolution of 200 m and thus, cannot resolve the near shore (wave breaking zone) sufficiently to reproduce wave setup. In addition, we calibrated the coastal ocean model by increasing the wind speed by 7%. This allowed us to minimize the error in predicting the peak water levels and, using this method, we consider missing processes such as wave formation, lack of resolution and errors in the atmospheric model. Yet, the reason for the underestimation of high events is not entirely clear. Despite the 7% increase of wind speed, the underestimation may be partially explained by the missing process of wave setup. Another reason could be that those few storms that were responsible for the underestimated surges are not reproduced well by the atmospheric data (Lorenz and Gräwe 2023). Nevertheless, the model GEV distribution overestimates the 200-year return levels compared to the tide gauges. Thus, we expect the uncertainty of wave setup on the flood maps to be small, especially for the 200-year cases with and without SLR.*

**4.13**

Line 396: . *Second, there is still no conclusive information on potential changes in wave climate, and results show strong spatio-temporal variability (Weisse et al., 2021).* I don't see why this is a reason not to include wave setup in your assessment, which is based on a hindcast and not projections. You are also not evaluating (future) changes in storm surge, but you still include this process.

We thank reviewer #2 again for raising this important point. Please see also our response to your comment 4.12. There are two main reasons for the exclusion of wave setup:

1. In the coastal ocean model, we have a resolution of 200 m. Even if we ran a wave model on the same grid, we would not resolve the wave breaking zone and the nearshore bathymetry (winter and summer sandbars (migration)). Here we assume that we need at least 3 grid points to resolve bathymetric features, so we fail to predict the correct wave height at the breaking point.
2. As wave models are much more computationally expensive than hydrodynamic models, one would run the wave model at a much coarser resolution. So, we would fail to resolve the nearshore wave dynamics.

However, we calibrated the wave model by changing the wind speed and increasing the wind by a constant factor. This allowed us to minimize the error in predicting the peak water levels. With this method we have already considered missing processes (wave formation, lack of resolution, errors in the atmospheric model, loading, ...).

In order to estimate a value of the potential errors we make by excluding wave effects, we use a simplified formula to estimate the wave setup. Here we use the formulation of Guza and Thornton (1981):

wavesetup = 0.16*significantwaveheight

Assuming a significant wave height of 1-1.5 m outside the breaking zone, we estimate a wave setup of 15-25 cm. However, these figures are variable, depending on e.g. the slope of the beach, the position of the sandbank and the reflection at the shoreline.

With respect to the projections: The extrapolation of 200-year return water levels are based on a hindcast, but we model and now focus our paper on SLR for the year 2100, which is actually a projection. How meteorological conditions may change and how waves and wave setup may be affected by higher sea levels, is not clear at this point (Weisse et al. 2021), which is another reason why we decided to leave waves out of the study.

Finally, while future changes in wave characteristics are not conclusive, studies suggest that future storm surges will not be affected by SLR, thus allowing to linearly add SLR to today´s storm surges (Gräwe and Burchard 2012; Hieronymus et al. 2018).

**4.14**

Line 397: *Finally, potential wave setup along the German Baltic Sea is arguably low compared to uncertainties associated with the simulated SLR projections.* Do you have a reference for this?

This sentence has been deleted in the new version of the manuscript. See the updated text in our response to comment 4.12.

**4.15**

Line 417: *associated* do you mean 'future'

Thanks. We have now clarified this sentence:

*Ignoring the potential of dike failure, we therefore expect that the increase in dike heights by 1.5 m could offset a fraction of the additional flood extent caused by the SLR-scenario.*

**4.16**

Some limitations haven't been discussed: Lack of processes in the hydrodynamic model (e.g. mean sea level variations), lack of resolution to solve some regions, source of general underestimation of extremes, the uncertainties in the EVA method (especially when looking at such high return periods, general rule of thumb is that records can be extrapolated via EV distributions up to 3 times their length). The hydrograph method must also introduce a lot of uncertainties, and I presume the flood maps are quite sensitive to the choice of threshold used

for the hydrograph design as well as to the duration of the event, now fixed at 6 days. How does this compare to the duration of the event during the January 2019 storm? How does the hydrograph vs time-series approach compare for this particular event, when you use the corresponding return period?

Thank you for your comment. We have clarified in section 3.4 that mean sea level variations are included in the coastal ocean model (GETM) and we have taken this into account by detrending the hindcast of water levels across the study region. Please see revised section 3.4 and also our response to your comment 3.5.

From revised section 3.4:

*Using this model, we extrapolate 200-year return water levels for each tide gauge and boundary station. Note that slow, long-term variations in mean sea level have been subtracted from the time series by a linear fit. Therefore, we only consider ESLs relative to mean sea level in the statistics.*

In the discussion, we have now added wave setup as a possible reason for the underestimation of ESL. A technical reason why we have not taken wave setup into account is that we are modeling at a resolution of 200 m, which is too coarse to resolve the near shore sufficiently well (see also text in our response to your comment 4.12).

We have added the hydrograph for the 2019 event as a grey line to former Figure A1, which has now been moved to the main text. This enables the reader to compare the duration and shape of the 2019 event with the hydrographs of the constructed 200-year events. We hope this clarifies your question.

We now also discuss potential effects of the hydrograph method on simulated flood extents in section 4.6. We have added the following lines:

*Only a few studies have examined the impact of surge duration and intensity on flood characteristics, but* Höffken et al. (2020) *have shown for a case study in the German Baltic Sea that flood extents can vary by 20 % when sea levels rise. The Baltic Sea is characterized by a microtidal regime, which means that high water levels during storm surges can stay for several days and various storm surge intensities are observed* (MacPherson et al. 2019). *Consequently, storm surge hydrographs are spatially and temporarily (between different storm surge events) variable. While we account for the spatial variability by calculating mean storm surge hydrographs for 32 flood boundary stations (Fig. 1) across the study region, the temporal variability is not accounted for as we apply mean surge shapes. Depending on the intensity of surges with a 200-year return water level, we may therefore both over- and underestimate simulated flood extents.*

**5. **Conclusions**

Modify according to the objective of the manuscript. If the objective is to present the framework, include other possible applications, and future directions of development based on the current assessment limitations and envisioned applications. If the objective was the 200RP flood characteristics dataset, explain how it can be used for spatial planning, for

(climate) dike design, etc. Some of these aspects are now briefly mentioned between lines 430 and 435 but without a clear objective stated for the paper at the beginning, it is difficult to understand.

Thanks for this remark. We have now rephrased the conclusions and refer to the updated objectives stated in our response to your comment 2.1.

**Publication bibliography**

Arns, Arne; Wahl, Thomas; Wolff, Claudia; Vafeidis, Athanasios T.; Haigh, Ivan D.; Woodworth, Philip et al. (2020): Non-linear interaction modulates global extreme sea levels, coastal flood exposure, and impacts. In *Nature communications* 11 (1), p. 1918. DOI: 10.1038/s41467-020-15752-5.

Gräwe, Ulf; Burchard, Hans (2012): Storm surges in the Western Baltic Sea: the present and a possible future. In *Clim Dyn* 39 (1-2), pp. 165–183. DOI: 10.1007/s00382-011-1185-z.

Guza, R. T.; Thornton, E. B. (1981): Wave set-up on a natural beach. In *J. Geophys. Res.* 86 (C5), p. 4133. DOI: 10.1029/JC086iC05p04133.

Hieronymus, Magnus; Dieterich, Christian; Andersson, Helén; Hordoir, Robinson (2018): The effects of mean sea level rise and strengthened winds on extreme sea levels in the Baltic Sea. In *Theoretical and Applied Mechanics Letters* 8 (6), pp. 366–371. DOI: 10.1016/j.taml.2018.06.008.

Höffken, Jorid; Vafeidis, Athanasios T.; MacPherson, Leigh R.; Dangendorf, Sönke (2020): Effects of the Temporal Variability of Storm Surges on Coastal Flooding. In *Front. Mar. Sci.* 7, Article 98. DOI: 10.3389/fmars.2020.00098.

Lorenz, Marvin; Gräwe, Ulf (2023): Uncertainties and discrepancies in the representation of recent storm surges in a non-tidal semi-enclosed basin: a hind-cast ensemble for the Baltic Sea.

MacPherson, Leigh R.; Arns, Arne; Dangendorf, Sönke; Vafeidis, Athanasios T.; Jensen, Jürgen (2019): A Stochastic Extreme Sea Level Model for the German Baltic Sea Coast. In *J. Geophys. Res. Oceans* 124 (3), pp. 2054–2071. DOI: 10.1029/2018JC014718.

Melund (2022): Generalplan Küstenschutz des Landes Schleswig-Holstein. Fortschreibung 2022. Kiel. Available online at https://www.schleswig-holstein.de/DE/fachinhalte/K/kuestenschutz/Downloads/Generalplan.pdf?__blob=publicationFile&v=2.

Muis, Sanne; Apecechea, Maialen Irazoqui; Dullaart, Job; Lima Rego, Joao de; Madsen, Kristine Skovgaard; Su, Jian et al. (2020): A High-Resolution Global Dataset of Extreme Sea Levels, Tides, and Storm Surges, Including Future Projections. In *Front. Mar. Sci.* 7, Article 263. DOI: 10.3389/fmars.2020.00263.

Schuldt, Caroline; Schiewe, Jochen; Kröger, Johannes (2020): Sea-Level Rise in Northern Germany: A GIS-Based Simulation and Visualization. In *KN J. Cartogr. Geogr. Inf.* 70 (4), pp. 145–154. DOI: 10.1007/s42489-020-00059-8.

Soomere, T.; Pindsoo, K.; Bishop, S. R.; Käärd, A.; Valdmann, A. (2013): Mapping wave set-up near a complex geometric urban coastline. In *Nat. Hazards Earth Syst. Sci.* 13 (11), pp. 3049–3061. DOI: 10.5194/nhess-13-3049-2013.

StALU (2012): Regelwerk Küstenschutz Mecklenburg-Vorpommern. Küstenraum und Bemessungsgrößen von Küstenschutzanlagen in M-V. With assistance of Knut Sommermeier. Rostock: Verlag Redieck & Schade GmbH.

Sterr, Horst (2008): Assessment of Vulnerability and Adaptation to Sea-Level Rise for the Coastal Zone of Germany. In *Journal of Coastal Research* 242, pp. 380–393. DOI: 10.2112/07A-0011.1.

van der Pol, Thomas; Hinkel, Jochen; Merkens, Jan; MacPherson, Leigh; Vafeidis, Athanasios T.; Arns, Arne; Dangendorf, Sönke (2021): Regional economic analysis of flood defence heights at the German Baltic Sea coast: A multi-method cost-benefit approach for flood prevention. In *Climate Risk Management* 32, p. 100289. DOI: 10.1016/j.crm.2021.100289.

Vollstedt, Bente; Koerth, Jana; Tsakiris, Maureen; Nieskens, Nora; Vafeidis, Athanasios T. (2021): Co-production of climate services: A story map for future coastal flooding for the city of Flensburg. In *Climate Services* 22, p. 100225. DOI: 10.1016/j.cliser.2021.100225.

Vousdoukas, Michalis I.; Bouziotas, Dimitrios; Giardino, Alessio; Bouwer, Laurens M.; Voukouvalas, Evangelos; Mentaschi, Lorenzo; Feyen, Luc (2018): Understanding epistemic uncertainty in large-scale coastal flood risk assessment for present and future climates. In *Nat. Hazards Earth Syst. Sci.* (18), pp. 2127–2142. DOI: 10.5194/nhess-18-2127-2018.

Weisse, Ralf; Dailidienė, Inga; Hünicke, Birgit; Kahma, Kimmo; Madsen, Kristine; Omstedt, Anders et al. (2021): Sea level dynamics and coastal erosion in the Baltic Sea region. In *Earth Syst. Dynam.* 12 (3), pp. 871–898. DOI: 10.5194/esd-12-871-2021.

---

## Author Comment (AC3)

Revisions Manuscript:

*"A new modelling framework for regional assessment of extreme sea levels and associated coastal flooding along the German Baltic Sea coast"*

by Kiesel, J.; Lorenz, M.; König, M.; Gräwe, U.; Vafeidis, A.T.

https://doi.org/10.5194/nhess-2022-275

**Answers to reviewer #3**

We would like to thank anonymous referee #3 for the interesting and helpful comments, which have helped us to improve the manuscript. In response to these comments, we have clarified the objectives of our paper and rephrased the research questions. In addition, we have added some detail to the methods section related to the extreme value analysis and better explain the importance for adjusting the elevation data for dikes. Please find our detailed responses to your comments below.

*In order to increase readability, we have written our answers in green. All citations of text from the new, revised version of the manuscript are written in italics.*

**Comments anonymous referee #3**

The manuscrit by Kiesel and co authors adresses the difficult problem of coastal flooding at regional scale along Germany Baltic Sea with multiple difficulties, i.e. data resolution, account for dykes, validation, cascade of uncertainties, etc. The authors propose a modelling framework to do do with a key ingredient being the use of mention images for validation of the assessment.

I believe that the results have the potential to be of interest for a wide audience. Yet several aspects remain unclear and I recommend major corrections before publication.

**1)** Multiple problems are tackled but with different degrees of achievement, and as reader we get lost about the main message. Is the main result the framework ? If so, a figure presenting the different steps would help a lot together with a discussion section dedicated to the limitations of each this step. If the main result is the use of satellite images, this should be reflected in the title as well in the introduction.

We thank R3 for this helpful comment. We agree that the objectives of our paper were not clearly outlined and have therefore, also in response to a comment 2.1 by Reviewer 2 (see our respective response document), rephrased our objectives.

We have adjusted the wording in the text and now state more clearly the objectives of our paper before we elaborate about the methods used. We have also rewritten our conclusions accordingly.

Please find our specific edits below:

First, we have added a sentence at the beginning of the paragraph in the introduction describing the difficulties and limitations of coastal inundation modelling:

*State-of-the-art coastal flood maps should consider oceanographic forcings, projected SLR, detailed topographic data on coastal morphology, including anthropogenic coastal protection measures such as dikes, and the effects of land cover on flood propagation.*

Second, we point out that studies that have used state-of-the-art hydrodynamic modelling (considering temporal evolution of the surge and the effects of surface roughness) to assess coastal flooding along the entire German Baltic Sea coast still missing.

*Along the German Baltic Sea coast, existing studies on coastal flooding have either used state-of-the-art hydrodynamic models, but cover only a small fraction of the study region* (Höffken et al. 2020; Vollstedt et al. 2021)*, or assess potential flood extents for the entire region, but rely on global topographic data sources and apply the bathtub approach* (Schuldt et al. 2020)*. In addition, the validation of produced flood extents is not provided.*

*There is a need to simulate coastal flooding on a regional scale, considering the limitations of large-scale coastal flood mapping mentioned before. This is particularly true for topographic data sources and the incorporation of coastal protection infrastructure, which constitute the main bottlenecks for the quality of coastal flood risk assessment (Vousdoukas et al. 2018).*

Finally, we have rephrased the objective of the paper, now emphasizing that we aim to assess coastal flooding in the study region for an event that is equivalent to existing design heights for state embankments:

*Here, we simulate coastal flooding along the German Baltic Sea coast for a storm surge that aligns with the design standard of state embankments in the region, i.e.\ the 200-year return water level. This study aims at: 1) exploring how flood extent may change until the end of the century, if existing dikes are not upgraded, by applying two high-end SLR scenarios (1 m and 1.5 m); 2) identifying hotspots of coastal flooding in the study region, and 3); evaluating the use of SAR-imagery for validating the simulated flood extents. To the knowledge of the authors, this study constitutes the first regional-scale assessment using a high resolution, fully validated, and offline-coupled hydrodynamic modelling framework that incorporates natural and anthropogenic flood barriers to assess extreme sea levels and associated coastal flooding along the German Baltic Sea coast.*

**2)** the pre-treatment of the data for extreme value analysis is unclear to me. The authors use the notation 'esl' which gives the impression that the authors work with total water level. However the authors also mention extreme surges. If this is the second option, how do the authors derive the total water level at the coast to force the flooding model? More speciffically how do the authors handle the convolution with tide? In addition do the authors use skew surge or the instantaneous surge?

Thank you for this important comment (also see response to reviewer 2, comment 1). Since the Baltic Sea is characterised by a microtidal regime, the effects of tide-surge interactions on total water levels can be neglected (Arns et al. 2020). In order to clarify, we have added the tidal amplitudes to section 2, which now reads:

*The Baltic Sea is characterised by a microtidal regime (tidal range varying between 0.1 m and 0.2 m (Sterr, 2008), low salinity, strong stratification, and anoxic conditions in many areas (Meier et al., 2022).*

In addition, we clarify in section 3.4 that return water levels correspond to surges only (equal to total water level), as tides are negligible in the study region. Please see our suggestion for revision:

*Due to the microtidal regime of the Baltic Sea, the derived return periods and water levels correspond only to the surge component, as tidal contributions to ESL (tide-surge interactions) is negligible* (Arns et al. 2020).

**3)** the analysis of fig. 4 shows some similarities of the envelope of uncertainties. Could the authors provide more details about its computation. Finally what surprises me is that there is quasi systematic underestimation of the empirical points except for the more extreme points which is overestimated. Not having this last point would change completely the analysis. Could the authors comment on that?

Thank you for this comment, please see our response to a similar comment of reviewer R#2 below. In black, R#2´s comment and in green, our response.

The empirical values in the distribution shown for a TG (Figure 4) show an increasing bias for increasing RPs, but the fit seems insensitive to this. Could you elaborate? This seems also evident from the 2019 event, what is the corresponding RP and how does the error compare to the derived RP error, given that a different meteo forcing is used? The mean (negative) bias for the 2019 event is partly compensated by the positive bias in the stations around some of the unresolved lagoons, you could leave these out to have a better picture of the model performance for this event? (since TGs are used for the flooding at these locations anyways)

Yes, the model generally has a negative bias, especially the for the high ESLs. However, by overestimating the highest event in the model time period, the statistical fits of the model and the observations are very close, see also the return levels in Tab. 4. For some tide gauge locations, the 200-year return levels from the model are higher than return levels obtained from the observation based GEV. This is especially true for the coast of Schleswig-Holstein, where one event is overestimated by the model, thus influencing the tail of the distribution. However, the 95% confidence intervals are large due to the extrapolation and both, the estimated mean return levels of the model and the observation fall into the confidence intervals of each other.

We have compared the observed/measured (TG) return periods for the 2019 event (blue numbers in Fig. 1) with the return periods calculated with our model (black in Fig. 1). We find that the modelled values are within the respective confidence interval for most locations.

However, the estimation of return periods based on the discrete observations (explanation of the term "discrete observation" provided in caption of Fig. 1) are much smaller for many locations compared to the respective RP from the GEV fits (red numbers for GEV based on observations, black numbers for GEV based on model results). We cannot compare the errors here to other meteorological forcings since we only have run this model for the UERRA dataset. But for the whole Baltic Sea, Lorenz und Gräwe (2023) show for a hindcast ensemble of 13 members at ~1.8 km resolution that the ensemble spread of annual mean maximum

water levels is in the order of the variability between the meteorological datasets for the 99[th] percentile wind speeds.

We prefer to not dismiss the lagoons from the comparison, but rather show the limitations of the model resolution.

[Figure]

Figure 1: Return levels and 95% confidence intervals based on the observation GEV (red), model GEV (black), and for the discrete return periods of the 2019 event. Discrete means that in case the 2019 event is the highest for a 50-year long data record, the RP is 50 years. If it would be the second highest, the discrete RP would be 25, and so on. The respective return periods from the GEV with a return level as high as the 2019 event are listed in the respective colors above. These RP are usually larger than the discrete ones.

**4)** The analysis of the differences with the image data (fig. 7 in particular) is of high interest. Could the authors elaborate more on the added value of their updated dem with dykes information. Would it make sense to also compare the results with the ones of a traditional approach without this information? In addition, would it be possible to plot some examples of dem in the vicinity of the dyke to picture how much correction should be done?

We thank Reviewer #3 for this comment. The incorporation of detailed information on the position and elevation of dikes in the study region constitutes one of the major improvements of the developed modeling framework as compared to many other regional or even continental scale assessments (Vousdoukas et al. 2018; Vousdoukas et al. 2016; Hinkel et al. 2021). In order to make this point clearer, we have added a sentence to the respective method section 3.2.3.

The text now reads:

*We incorporated detailed information on dikes (location and height) in the modeling framework by using a high-resolution LiDAR derived (1 x 1 m) DEM and comprehensive datasets on the location of both state and regional dikes from local state authorities (Table 1). The incorporation of dikes constitutes one of the major improvements of the applied modeling framework as compared to previous regional or continental scale assessments (Vousdoukas et al. 2016). Elevation data and coastal protection levels are considered as the main bottlenecks for the quality of coastal flood risk assessments (Vousdoukas et al. 2018; Hinkel et al. 2021). Without correcting for dike heights in a 50 m DEM, the simulated flood extent will be overestimated, as the elevation of the dike heights are averaged out due to the resolution (Vousdoukas et al. 2012a; Vousdoukas et al. 2012b). The difference of a DEM with and without dike height correction is shown in Appendix Fig. A1 (here Figure 2).*

In order to clarify the importance of the conducted dike height corrections, we further added a figure to the appendix (now Fig. A1), in which we compare the corrected and uncorrected dike height elevations, as suggested by R#3.

[Figure]

*Figure 2: The difference between a DEM that was corrected for dikes (panels b) and d)) and an uncorrected DEM. A) and b) depict the area and dikes around Zingst (MP) and panels c) and d) show the northwestern coastline of the island of Fehmarn (SH).*

**5)** the approach described in appendix A is interesting to account for the time evolution around the surge peak. Given the variability shown in panel a) of fig. A1, could the authors comment on the possible impact of using only the mean time signal instead of a model accounting for this variability?

Thank you for this comment. In response, we have now moved the method from Appendix A1 into the main body of the text. We agree that varying surge duration/intensities can affect coastal flood characteristics. Therefore, we have extended our discussion in section 4.6. The text now reads:

*Only a few studies have examined the impact of surge duration and intensity on flood characteristics, but* Höffken et al. (2020) *have shown for a case study in the German Baltic Sea that flood extents can vary by 20 % when sea levels rise. The Baltic Sea is characterized by a microtidal regime, which means that high water levels during storm surges can stay for several days and various storm surge intensities are observed* (MacPherson et al. 2019). *Consequently, storm surge hydrographs are spatially and temporarily (between different storm surge events) variable. While we account for the spatial variability by calculating mean storm surge hydrographs for 32 flood boundary stations (Fig. 1) across the study region, the temporal variability is not accounted for as we apply mean surge shapes. Depending on the intensity of surges with a 200-year return water level, we may therefore both over- and underestimate simulated flood extents.*

**References**

Arns, Arne; Wahl, Thomas; Wolff, Claudia; Vafeidis, Athanasios T.; Haigh, Ivan D.; Woodworth, Philip et al. (2020): Non-linear interaction modulates global extreme sea levels, coastal flood exposure, and impacts. In: *Nature communications* 11 (1), S. 1918. DOI: 10.1038/s41467-020-15752-5.

Hinkel, J.; Feyen, L.; Hemer, M.; Le Cozannet, G.; Lincke, D.; Marcos, M. et al. (2021): Uncertainty and Bias in Global to Regional Scale Assessments of Current and Future Coastal Flood Risk. In: *Earth's Future* 9 (7), e2020EF001882. DOI: 10.1029/2020EF001882.

Höffken, Jorid; Vafeidis, Athanasios T.; MacPherson, Leigh R.; Dangendorf, Sönke (2020): Effects of the Temporal Variability of Storm Surges on Coastal Flooding. In: *Front. Mar. Sci.* 7, Artikel 98. DOI: 10.3389/fmars.2020.00098.

Lorenz, Marvin; Gräwe, Ulf (2023): Uncertainties and discrepancies in the representation of recent storm surges in a non-tidal semi-enclosed basin: a hind-cast ensemble for the Baltic Sea.

Schuldt, Caroline; Schiewe, Jochen; Kröger, Johannes (2020): Sea-Level Rise in Northern Germany: A GIS-Based Simulation and Visualization. In: *KN J. Cartogr. Geogr. Inf.* 70 (4), S. 145–154. DOI: 10.1007/s42489-020-00059-8.

Vollstedt, Bente; Koerth, Jana; Tsakiris, Maureen; Nieskens, Nora; Vafeidis, Athanasios T. (2021): Co-production of climate services: A story map for future coastal flooding for the city of Flensburg. In: *Climate Services* 22, S. 100225. DOI: 10.1016/j.cliser.2021.100225.

Vousdoukas, Michalis I.; Bouziotas, Dimitrios; Giardino, Alessio; Bouwer, Laurens M.; Voukouvalas, Evangelos; Mentaschi, Lorenzo; Feyen, Luc (2018): Understanding epistemic uncertainty in large-scale coastal flood risk assessment for present and future climates. In: *Nat. Hazards Earth Syst. Sci.* (18), S. 2127–2142. DOI: 10.5194/nhess-18-2127-2018.

Vousdoukas, Michalis I.; Ferreira, Óscar; Almeida, Luís P.; Pacheco, André (2012a): Toward reliable storm-hazard forecasts: XBeach calibration and its potential application in an

operational early-warning system. In: *Ocean Dynamics* 62 (7), S. 1001–1015. DOI: 10.1007/s10236-012-0544-6.

Vousdoukas, Michalis I.; Voukouvalas, Evangelos; Mentaschi, Lorenzo; Dottori, Francesco; Giardino, Alessio; Bouziotas, Dimitrios et al. (2016): Developments in large-scale coastal flood hazard mapping. In: *Nat. Hazards Earth Syst. Sci.* 16 (8), S. 1841–1853. DOI: 10.5194/nhess-16-1841-2016.

Vousdoukas, Michalis Ioannis; Wziatek, Dagmara; Almeida, Luis Pedro (2012b): Coastal vulnerability assessment based on video wave run-up observations at a mesotidal, steep-sloped beach. In: *Ocean Dynamics* 62 (1), S. 123–137. DOI: 10.1007/s10236-011-0480-x.